

# An Evaluation and Implementation of the Regional Coupled Ice-Ocean Model of the Baltic Sea.

Jaromir Jakacki and Sebastian Meler

Institute of Oceanology of Polish Academy of Sciences

*Correspondence to:* Jaromir jakacki (jjakacki@iopan.gda.pl)

**Abstract.** A three dimensional, regional coupled ice-ocean model based on the open-source Community Earth System Model has been developed and implemented for the Baltic Sea. The model consists of 66 vertical levels and has a horizontal resolution of approx. 2.3 km. The paper focuses on sea ice component results but the main changes have been introduced in the ocean part of the coupled model. The hydrodynamic part, being one of the most important components, has been also presented and validated. The ice model results were validated against the radar and satellite data, and the method of validation based on probability was introduced. In the last two decades satellite and model results show an increase in the ice extent over the whole Baltic Sea, which is an evidence of a negative trend in air temperature in recent decades and increasing of winter discharge from the catchment area.

## 1 Introduction

In the Baltic Sea coupled ice-ocean models have been developed and used for more than two decades, whereas coupled land-atmosphere-sea models - for about 15 years [BACC and BACC II, (Omstedt et al., 2004, 2014)]. First high-resolution fully coupled atmosphere-sea-ice-ocean-land-surface models (Döscher et al. (2002); Gustafsson et al. (1998); Hagedorn et al. (2000); Schrum et al. (2003)) were built during BALTEX Phase I. The models were developed to improve short-range weather forecast. In the beginning of BALTEX Phase II two fully coupled atmosphere-sea-ice-ocean models were also available (Döscher et al. (2002); Lehmann et al. (2004)). Now other groups are developing their own models (e.g., Dieterich et al. (2013); Pham et al. (2014); Tian & Boberg (2013) of the Baltic Sea region. The second phase of BALTEX focused more on studying climate change. Coupled atmosphere-sea-ice-ocean models were further elaborated using a hierarchy of sub-models of the earth system combining RCMs (Regional Climate Models) with sub-models of surface waves (Almroth Rosell et al. (2016); Rutgersson et al. (2012)), land vegetation (Smith et al., 2011), hydrology and land biochemistry (Arheime et.al (2012); Mörth et al. (2007)), marine biogeochemistry (Daewel & Schrum (2013); Eilola et. al. (2009, 2011); Neumann et al. (2002); Savchuk et al. (2012)), the marine carbon cycle (Edman & Omstedt (2013); Gustafsson et al. (2014); Kuznetsov & Neumann (2013); Omstedt et al. (2010)), marine biology (Hense et al. (2013); Meier et al. (2011)), and food web modelling (MacKenzie et al. (2012); Niiranen et al. (2013)) as well as with socioeconomic impact assessments (Piwowarczyk et al., 2012). The Gulf of Finland was simulated using a horizontal resolution of 0.25 nautical miles (Andrejev et.al, 2010). Gräwe et al. (2013) studied saltwater inflows in the future climate using a model of the western Baltic Sea with a horizontal resolution of less than 1 km. River



inflows are represented by large-scale HBV hydrological model (Lindström et al., 1997) of the entire Baltic catchment area (Graham, 1999, 2004). Later, two new hydrological models have been developed to calculate future river flows and river-borne nutrient loadings, i.e. the Hydrological Predictions for the Environment (HYPE) model ((Arheime et.al, 2012; Lindström et al., 2010)) and the Catchment Simulation Model (CSIM) (Mörth et al., 2007). As meteorological forcing, high-resolution Baltic

Sea simulations typically use the output of local atmosphere models, such as different versions of the High Resolution Limited Area Model (HIRLAM) (www.hirlam.org) or the Deutscher Wetterdienst (DWD) model (www.dwd.de), or ERA-40 reanalysis data (Uppala et al., 2005) or their downscalings (Höglund e al., 2009; Samuelsson et al., 2011).

In this work, the recently developed Community Earth System Model (CESM, Hurrell et al. (2013) has been used and implemented for the whole Baltic Sea. CESM is a descendant of the Community Climate Model (CCM, Williamson et al. (1987);

Hack et al. (1993)), the Climate System Model (CSM) and the Community Climate System Model (CCSM version 3, Blackmon et al. (2001) developed by the National Centre for Atmospheric Research (NCAR). Up to the mid-1990s, the CCM was of limited use as a climate model because it did not include submodels of the global ocean and sea ice.

The first plan to develop and use a climate system model (CSM) including the atmosphere, land surface, ocean, and sea ice was proposed to the National Science Foundation (NSF) in 1994 (Blackmon et al. , 2001) The recent releases were also im-

proved by the University Centre for Atmospheric Research (UCAR). The latest CCSM version 4 consists of four separate components, atmosphere (atm), ocean (ocn), sea-ice (ice) and land (lnd), coupled by a central coupler component. The main difference between current version of CCSM and CESM is that new biogeochemistry capabilities were added in the CESM model. It means that CESM is only a newer version of CCSM. It is also important, that CESM includes CCSM – it is possible to run simulations implemented in CCSM using CESM model. CESM has a set of configurations and selected ones have been

already scientifically validated. All configurations are based on global grids, except the RACM/RASM project (**R**egional **A**rctic **C**limate/ **S**ystem **M**odel) that develops all active components in a coupled, regional Pan-Arctic climate model ((Maslowski et al., 2009), (Cassano et. al., 2010), (Maslowski et al., 2012)). Since resolutions of the provided configurations are not suitable for the Baltic Sea, one of challenges in the presented implementation was to adequately increase both horizontal and vertical resolution.

For the purpose of this study the CESM was adopted for the Baltic Sea. The implemented version includes two active components, ocean (**P**arallel **O**cean **P**rogram, POP) and ice (**C**ommunity **I**ce **C**od**E**, CICE), and two data models representing atmosphere (datm) and land (dlnd). The active (dynamical) components are fully prognostic models, whereas the basic function of data models is reading external data (interpolated outside of the model), modifying that data and sending them to the central coupler. The central coupler (cpl7) is responsible for the coordination and the flow of information between all compo-

nents. The coupler and other models have no fundamental knowledge of whether another component is fully active or just a data model. The implemented setup of the Baltic Sea model is shown in Fig. 1. We call this model B-CESM.

In the above configuration, different data sets have been used as external forcings. The river runoff data from the **Balt**ic **HY**drological **P**redictions for the **E**nvironment model (Balt-Hype, (Lindström et al., 2010)), covering the years 1971-2008 were provided by the Swedish Meteorological and Hydrological Institute (SMHI). The river runoff climatology was calculated

from available period and employed in the model for missing periods. In total, data for 75 rivers have been implemented in



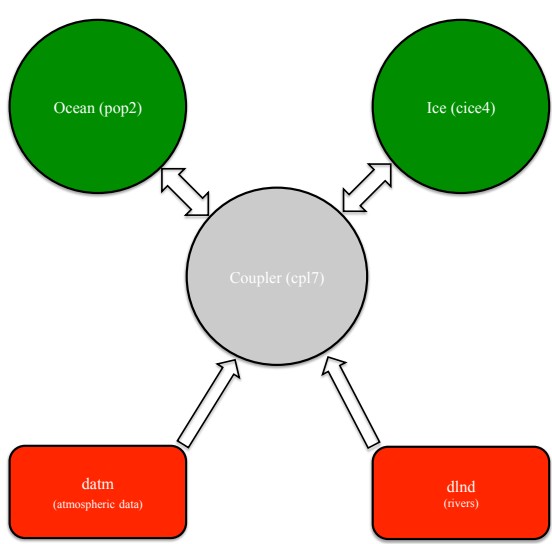

**Figure 1.** The Baltic Sea coupled model diagram

the model. Two major methods are commonly used for implementing the river runoff data in ocean models. The most often applied approach is to treat a river runoff as precipitation, which changes salinity of the uppermost cells. Another way is to add the freshwater input as precipitation but instead salinity change, the freshwater volume is added to the model upper cells. This requires removal of the additional water volume at the lateral boundary of a regional model as was done in the presented Baltic

Sea model. Our approach is non-standard and has been implemented only for the purpose of this model for the Vistula River. A special channel has been added to the model domain and precipitation has been released only within this channel. The sea level difference between the channel and the Gulf of Gdansk assured velocity of the freshwater flow, which could not be obtained, if the river runoff was implemented as precipitation over the sea. The implemented channel approach gives much better results for large rivers, such as Vistula or Lena, while for smaller rivers including the river runoff as precipitation is commonly used.

The presented results were obtained with the Baltic Sea model using atmospheric forcing from ERA Interim reanalysis (ERAi, ECMWF, 1979-present, 0.75 degree horizontal resolution, (Berrisford et al. , 2011)) and data from the local forecast model UM run by the Interdisciplinary Modeling Centre at the University of Warsaw (ICM, approx. 4 km horizontal resolution, 2010-2014). Atmospheric forcing variables required by the model includes:

– temperature at 2 meters,

– specific humidity at 2 meters,

– wind speed and direction at 10 meters,



- mean sea level pressure,

- short and long wave radiation downward,

- total snow and rain precipitation.

The availability of global atmospheric reanalyses permits to use them from different data sources. For example in the case of

NCEP (National Centers for Environmental Prediction) (Kalnay, 1996; Saha et al., 2010), JMA (Japan Meteorological Agency) (Onogi et al., 2007), NASA (National Aeronautics and Space Administration) (Rienecker et al., 2011; Schubert et al., 1993), and ECMWF ((Dee et al. , 2011; Berrisford et al. , 2011), it is possible to use data that cover the Baltic Sea area and all of them have resolution lower then one degree. Also regional analysis made by High Resolution Limited-Area Model (HIRLAM) are provided with 22 km horizontal resolution. This reanalysis has been done using ERAi data as boundary conditions. In our

case we use ERAi because it covers time of integration (which is limited by observational sea level data for Goteborg station) and runoff data are available for the same period. Also data from ECMWF could be used in the Baltic Sea modeling with reasonable accuracy (Omstedt &Wesslander , 2005).

All models (components) are governed by the central coupler (cpl7), based on the **M**odel **C**oupling **T**oolkit (MCT). It controls the exchange of data between the individual models and all time steps. The coupling time step for all models has been set to

600 seconds and the same value was also used as the ice model time step. The ocean model has its own system that controls its time steps, which are not uniform. The requested time step of the ocean model has been equal to 150 seconds and we can assume that it has been the longest step of the POP.

## 2  Ocean model description and adaptation to the Baltic Sea

The ocean component of the Baltic Sea model in the presented configuration is the Parallel Ocean Program, a well-known,

3-dimensional, z-coordinate general circulation model ((Smith & Gent, 2004)). The model uses hydrostatic and Boussinesq approximations for the primitive equations of the fluid motion on the orthogonal spherical grid. A free surface is implemented as pressure at the top boundary layer. A detailed description of the POP model and improvements can be found in numerous papers, e.g. (Bryan, 1969; Semtner, 1974; Cox , 1984; Killworth et al., 1991; Mintz & Semtner, 1977).

### 2.1  Initial state of the model

Both active models (ice and ocean) adopt a rotated spherical grid with horizontal resolution of 1/48 degree (approx. 2.3 km). To obtain the same size of all cells in a horizontal plane, the equator of the grid is located in the center of the domain. The model bathymetry is based on the ETOPO1 (Earth Topography Digital Dataset), a global 1 arc-minute relief model of the Earth's surface (Amante& Eakins , 2009). The B-CESM has 66 vertical levels, of which the first 50 levels are 5 m thick, whereas thickness increases in deeper layers. The number of levels has been chosen to optimize the computational cost in the model,

at the same time covering most of the Baltic Sea with identical cells. The compromise resulted in 66 vertical levels shown in Fig. 2 together with the model domain and bathymetry (colors represent vertical levels).

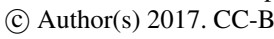



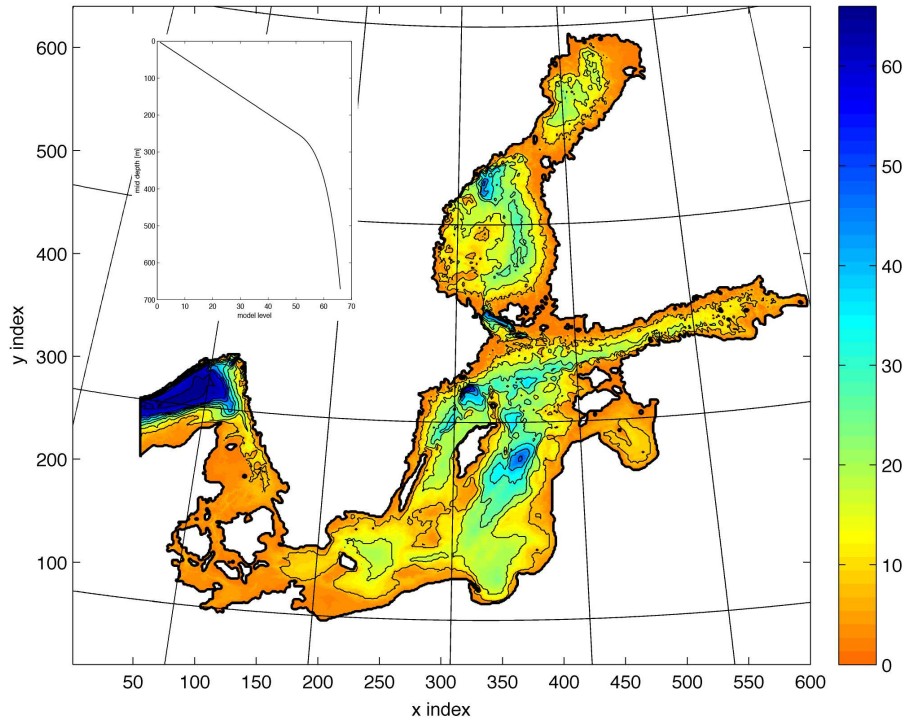

**Figure 2.** Model domain, bathymetry (color scale in model levels) and vertical resolution (insert).

Initial conditions have been determined based on data provided by Copernicus (Marine Environment Monitoring Service http://marine.copernicus.eu), a global monitoring and analyzing community. For the purpose of this model physical reanalysis, included 3D-var data assimilation, based on High Resolution Model of the Baltic Sea (HIROMB) model has been applied. Swedish Meteorological and Hydrological Institute (SMHI) has been doing the reanalysis.

5 Initial temperature and salinity fields for January 1990 were interpolated over the model domain. The model started from no-motion state, which is also called a 'cold start'. The model was spun up for three years (1990-1992) and then went back to 1 January 1990. In the Baltic Sea strong autumn storms drive very deep turbulent mixing, therefore three years seems enough for the ocean model spin-up. In case of ice cover or heat balance, memory is closed to one year. The salinity memory of the Baltic Sea is over 30 years (Omstedt & Hansson, 2006). But in case of given observed (or reanalyzed) salinity, could be realistically 10 modeled for the whole Baltic Sea.

## 2.2 Lateral boundary conditions and their verification

Although the Baltic Sea is a semi-enclosed sea, a strong salinity difference between the North Sea and the Baltic Proper results in a difference in sea level of the order of 10 cm between the Kattegat and the South-West Baltic. Being a transit area, the Danish Straits play the key role in water exchange between the North Sea and the Baltic Sea. Therefore a proper implementation of



boundary conditions has strong influence upon the barotropic balance between the North Sea and the Baltic Sea and also upon the salinity distribution in the Baltic Sea. Two different approaches were used to implement lateral boundary conditions:

   a) sea level measured in Göteborg was adapted as a boundary condition in Kattegat,

   b) the Orlanski open boundary conditions (OBC) were enforced at the boundary area.

Located in the narrow and shallow connection between Skagerrak and Kattegat, Göteborg provides a convenient point for applying boundary conditions. Moreover, sea level data for this location are openly available for over twenty years (1990-today). To apply sea level assimilation for the Goteborg area, it has been necessary to introduce a modification in the barotropic equation, originally formulated as (Dukowicz & Smith , 1994):

$$\begin{cases} \partial_t u - fv = -g\partial_x \eta + G^x \\ \partial_t v + fu = -g\partial_y \eta + G^y \\ \partial_t \eta + \partial_x Hu + \partial_y Hv = 0 \end{cases} \quad (1)$$

where $H$ is the total depth, $f$ is the Coriolis parameter, $(u,v)$ are the barotropic (vertically averaged) velocities, $g$ is the gravity acceleration, $G^x$ and $G^y$ are external forcings and $\eta$ is the surface pressure. Symbols $\partial_t, \partial_x, \partial_y$ represent time and space derivatives. The RHS of Eq.(1) represents x and y gradient components of the surface pressure. This part of the barotropic equation has been modified to incorporate sea surface height (SSH) at the boundary area by adding the gradient of difference between the model sea level and the SSH measured in Göteborg. The Cressman analysis scheme (Cressman, 1959), which

defines the weighting function in terms of a radius of influence, is widely used for simple assimilation systems. In our case the Hamming window ($w_h(x,y)$), which is very often used in signal processing, has been used instead of the Cressman weighting function. An advantage of this approach is that the Hamming window is represented by simple harmonic function that is the basic solution of the wave equation. The modified barotropic Eq.(1) has now the form:

$$\begin{cases} \partial_t u - fv = -g\partial_x \left( \eta + w\eta' \right) + G^x \\ \partial_t v + fu = -g\partial_y \left( \eta + w\eta' \right) + G^y \\ \quad \partial_t \eta + \partial_x Hu + \partial_y Hv = 0 \end{cases} \quad (2)$$

where $\eta'$ is the difference between the modeled and measured sea level in Göteborg.

The Sommerfeld radiation conditions (Sommerfeld, 1949) are most commonly used at the boundary. In our case the condition at the northern boundary is formulated as:

$$\frac{\partial \psi}{\partial t} + c\frac{\partial \psi}{\partial y} = 0 \quad (3)$$

Where $\psi$ is velocity, barotropic component of velocity or sea surface height at the lateral boundary in the model, and c is the

wave speed. Two approaches are often applied to find the c parameter. One takes into account the barotropic wave speed, which is constant and depends on the depth and gravitational acceleration. The second approach, introduced by Orlanski (1976), was implemented also in our model. It uses data from the previous time step to calculate phase speed, which is applied in the current





time step. Taking into account the varying $c$, Eq.(3) provides a prognostic variable at the boundary (Kantha & Clayson, 2000):

$$\psi_{jM}^{n+1} = \psi_{jM}^n - R\left(\psi_{jM}^n - \psi_{jM-1}^n\right) \tag{4}$$

where

$$R = \frac{c\Delta t}{\Delta x} = -\frac{\psi_{jM-1}^{n+1} - \psi_{jM-1}^n}{\psi_{jM-2}^n - \psi_{jM-2}^n} \tag{5}$$

In Eq.( 4 ) and ( 5 ) $n$ represents time step and $jM$ is the northern boundary index. In the B-CESM model Eq.(4) and (5) have

been applied in Kattegat.

Comparison of the sea level measured in Göteborg and obtained from the model after implementing the modified boundary

condition is shown in Fig. 3. Even if the modeled sea level is lower than the measured one, the applied modification of the

barotropic equation produces realistic sea level variability. The main reason of discrepancy in the sea level amplitude is pressure

averaging, turned on in the POP model to avoid the model becoming unstable. Pressure averaging allows for longer time step

in the ocean model and reasonable integration time.

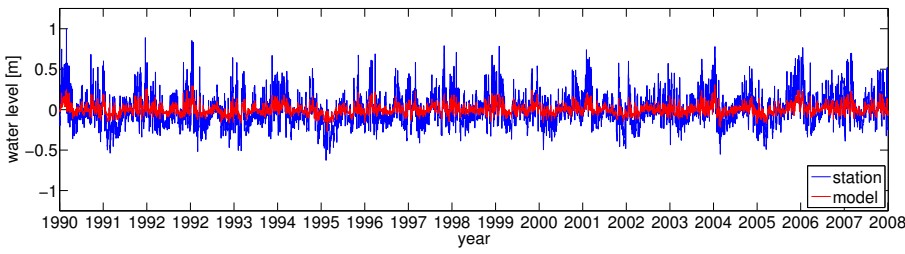

**Figure 3.** Modeled (red) and measured (blue) sea level in Göteborg.

Because the North Sea is the source of salt in the Baltic Sea, the flow through the Danish Straits is the most important driver

of the salt exchange between Kattegat and the South-West Baltic. The two main straits, Sund and Langeland Belt, are the main

passages for the salt transport into the Baltic (in example Lehmann et al. (2004); Mohrholz et al. (2015)). The flow through

Sund is mostly driven by the sea level pressure difference between the southern and northern entrances, and can be calculated

from the semi-empirical expression:

$$\Delta\eta = K_f \cdot Q \cdot |Q| \tag{6}$$

where $\Delta\eta$ is the sea level difference between the ends of Sund (Skanor and Hornbaek), $K_f$ is specific resistance, and $Q$ is the

flow through the strait.

The long-term sea level measurements are available for two stations shown in Fig. 4. Based on measurements, the specific

resistance in Sund was estimated on approximately $2.03 \times 10^{-10}$. Figure 4 shows that flow rate depends on the sea level

difference for the Sund. The black line represents measurements and the red points are from the model described herein. The

nonlinear regression provides the modeled specific resistance $Kf = 4.05e-10$. For comparison GETM (Grode, 2004), NB03



(the North Sea 3 Nm, (Funkquist & Kleine, 2007)) and BS01 (the Baltic Sea 1 Nm, (Funkquist & Kleine, 2007)) models provide $K_f$ value of $3.06e-10$, $3.78e-10$ and $12.5e-10$, respectively.

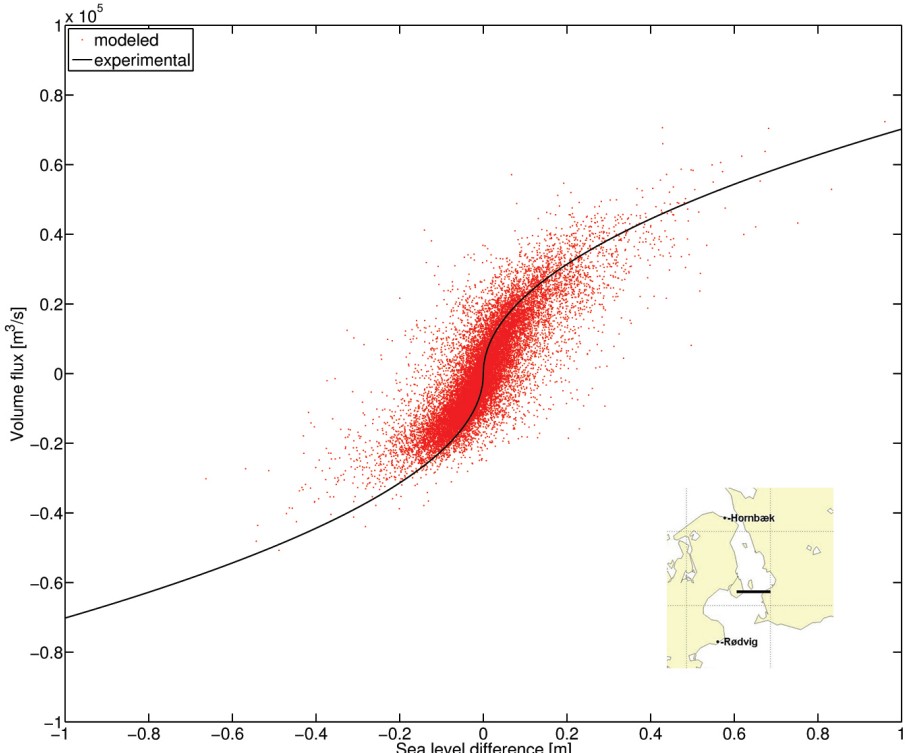

**Figure 4.** Flow rate vs. sea level difference in the Sund (red points – modeled, black line - calculated from the measured sea level difference). The insert shows position of two stations at the northern and southern end of Sund and the section.

The dynamical state of the Baltic Sea is the result of balance between dense and salty waters incoming through the Danish Straits and the sea level driven by winds, salinity difference, precipitation and rivers. Because of it, the annually averaged sea

5   level is expected to increase from the Danish Straits towards the Gulf of Bothnia . Figure 5 shows adjusted annual mean sea level at seven stations, located along the Baltic coast from west to east, obtained from three different numerical models: BS01, NB03 and GETM. BS01 and NB03 are the implementations of the HIROMB model with horizontal resolution of 1 and 3 nm respectively. GETM is a model set up by the Royal Danish Administration of Navigation and Hydrography and in this case it has a resolution of 1 nm (Grode, 2004).

10  The HIROMB models use meteorological forcing from HIRLAM (High Resolution Local Area Modelling, SMHI), while the GETM model is also driven by data obtained from HIRLAM, but from the Danish Meteorological Institute (DMI). The monitoring data were obtained from the height reference systems: Baltic HRS and Nordic Height System (NH60), which are 15 years long time series transformed from the Swedish Height System RH70 ((Carlsson, 1997)). Figure 5 shows the good agreement of the sea level spatial variations obtained from the presented model with HIROMB and GETM results, as well as





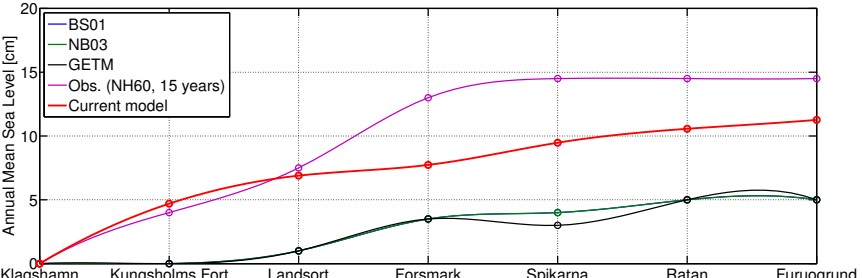

**Figure 5.** Comparison of the adjusted annual mean sea level from three different models, observations and current Baltic Sea model.

with observations. The averaged sea level varies between the individual models because of the different atmospheric forcing (in this case we use data from the ERA40 interim, ERA40 and Interdisciplinary Modeling Centre (ICM) at the University of Gdansk) and different resolutions. Additionally, pressure averaging in time has been implemented in the other models, resulting in smaller amplitude of the sea level variations.

## 2.3 Ocean model upper boundary layer

The conservation of heat at the top layer of the open ocean model requires (Yu at al, 2007):

$$Q_{net} = Q_{sw} + Q_{lw} + Q_{lat} + Q_{sen} \tag{7}$$

where subscripts $net$, $sw$, $lw$, $lat$, $sen$ indicate net heat, solar short waves radiation, long wave radiation, latent and sensible heat flux, respectively. It has been assumed that fluxes are directed to the ocean model (the ice model will be described separately). In our model, short and long wave radiations are provided by datm model as external forcing (see Fig. 1). Latent and sensible heat fluxes are sent to the active components and are calculated by the central coupler based on the bulk formula:

$$(E, H) = \rho_A |\Delta v| (C_E \Delta q, C_p C_H \Delta \theta) \tag{8}$$

where $E$ and $H$ are latent and sensible heat fluxes, $\rho_A$ is the air density, $\Delta v$ is the velocity difference between the ocean surface and atmosphere, $\Delta q$ and $\Delta \theta$ are specific humidity and temperature differences (between the atmosphere and the ocean surface) and $Cp$ is the specific heat capacity. $C_E$ and $C_H$ are transfer coefficients between the ocean surface and atmosphere (sensible heat transfer coefficient is the Stanton number, the latent heat coefficient is the Dalton number), and they depend on stability:

$$C_{(E,H)} = \kappa^2 \left[ \ln \left( \frac{Z_A}{Z_{0m}} \right) - \psi_m \right]^{-1} \left[ \ln \frac{Z_A}{Z_{0m}} - \psi_h \right]^{-1} \tag{9}$$

where $\psi_m$ and $\psi_h$ are the integration forms of the Monin-Obukhov similarity functions, $Z_A/Z_{0m}$ represents stability parameter, $Z_{0m}$ is the roughness length.

Wind – natural motion of the air – always exists over the ocean. Wind blowing over the ocean surface produces tangential

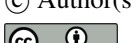



stress at the ocean-atmosphere interface, represented by a vertical flux of the horizontal momentum. Similar to heat fluxes, the horizontal momentum flux (wind stress $\tau_o$) can be also described by a bulk formula:

$$\tau_o = \rho_A C_d S \left( \mathbf{u}_A - \mathbf{u}_S \right) \tag{10}$$

where $C_d$ is drag coefficient, $S$ is the wind speed relative to the ocean surface and $\mathbf{u}_A$ and $\mathbf{u}_S$ are wind and ocean surface

velocities, respectively (vector quantities indicated by bold characters).

In consequence, the vertical shear is maximal at the surface of the ocean and decreases with depth. The interaction between atmosphere and ocean produces a mixed layer, occupying the upper part of the ocean, where temperature and salinity can be approximated as constant. The mixed layer depth (MLD) depends mostly on wind speed and direction, as well as its temporal and spatial scales and in shallow waters also on bottom topography. An important part of the energy transfer between

10 atmosphere and ocean upper layers is through turbulences. In the presented B-CESM model k-profile parameterization (kpp) was used (Large et al., 1994).

The most natural turbulence model is standard $k - \varepsilon$ model (Svensson, 1978; Rodi , 1980). Burchard & Petersen (1999) shown that two models: $k - \varepsilon$ and Mellor-Yamada level 2.5 (MY) (Mellor & Yamada , 1974, 1982; Li et al., 2001) perform similarily. Due to the fact that these schemes are not available in the POP model (MY is for sigma coordinate models only), kpp has

15 been modified to be more similar to MY based on Durski et al. (2004). The comparison of the two different schemes is not straightforward and to obtain a similar solution to MY, the formulation of interior shear was slightly modified in kpp. The modification was introduced in the relationship between the Richardson number and the vertical viscosity coefficient. The comparison between original and modified turbulent viscosity is shown in Fig. 6. Both solutions differ in the rate at which vertical mixing decays in the presence of stratification.





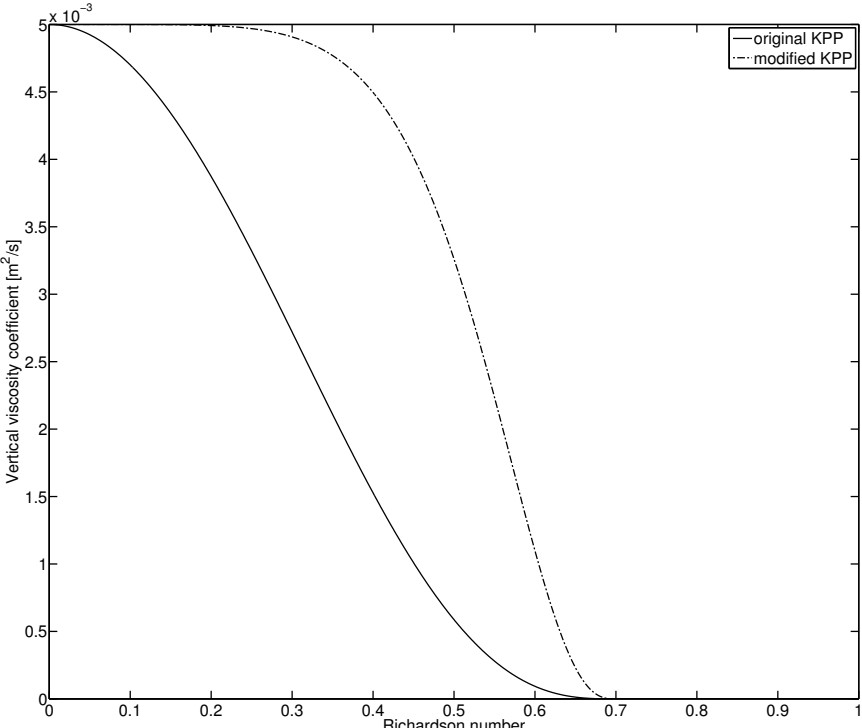

**Figure 6.** Original and modified (used in this simulations) turbulent viscosity (after Durski et al. (2004)).

## 2.4 Validation of the ocean model

Since the main focus of the current study is on the ice model, only selected results of the ocean model validation are presented in this paper. Figures 7 and present the comparison of the modeled sea surface temperature with observational and satellite data from different sources. The selected locations were determined by available buoys data, yet, unfortunately, no valid buoy data

5 exist for the Sodra Ostersjon. In this case only the modeled and satellite SST data are presented (in Fig. 8, in the upper right image).

All data except the model results are incomplete. The sea temperature from surface buoys was measured at the depth of 0.5 m. Satellite measurements of sea surface temperature (SST) are based on observations from up to 10 different satellites that measure in both infrared and microwave range. The satellite data have been obtained from the Ocean and Sea Ice Satellite

10 Application Facility (www.osi-saf.org) and the Group for High-Resolution Sea Surface Temperature (www.ghrsst.org). The SST data have been acquired from MODIS (Moderate Resolution Imaging Spectroradiometer) and they are based on the long wave SST (wavelengths are 11 and 12 micrometers). SST4 is four microns sea surface temperature (short wave SST). NSST is obtained only from night-time measurements.

Figure 8 shows a good agreement between observational data and model results. The model provides average temperature in



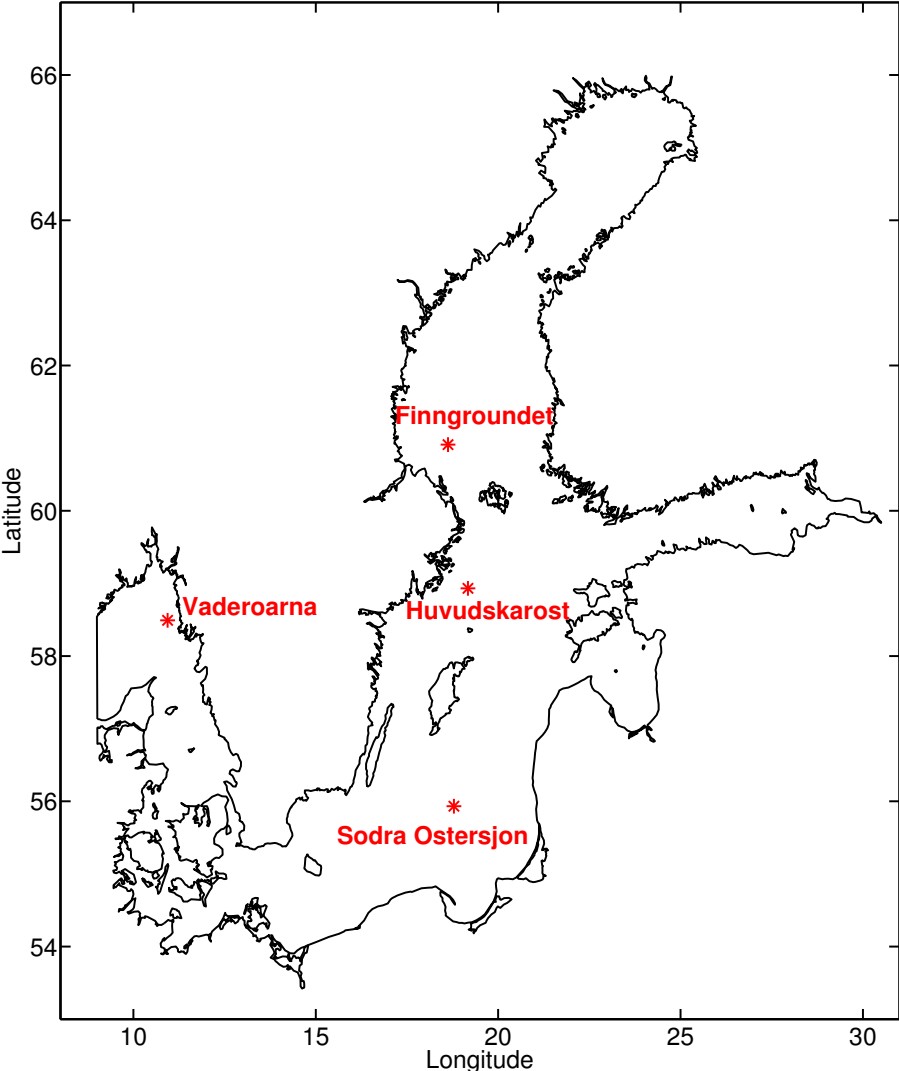

**Figure 7.** Locations of the buoys used for the comparison of the sea surface temperature.

the upper 5-m thick layer, whereas the satellite data represent sea temperature at the surface (usually the skin temperature). Therefore, the modeled sea surface temperature is lower than the measured one in summer and higher in winter. Validation of the modeled SST against satellite data is complemented by a comparison of one-year long time series of temperature and salinity profiles from the HuvudskarOst station (location shown on Fig. 1). Despite numerous gaps in in-situ measurements,

5   they can be still used for validation of the model results in terms of the vertical distribution of temperature and salinity and their seasonal variability. Fig. 9 shows that model satisfactorily reproduces temporal evolution of vertical profiles of temperature and salinity. Stratification, represented by depths of individual isotherms and isohalines is similar in observations and model results, and seasonal variability of both properties in the upper layer (there is not enough observational data in the deeper layers) is

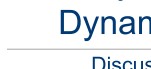
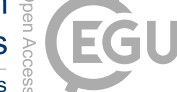

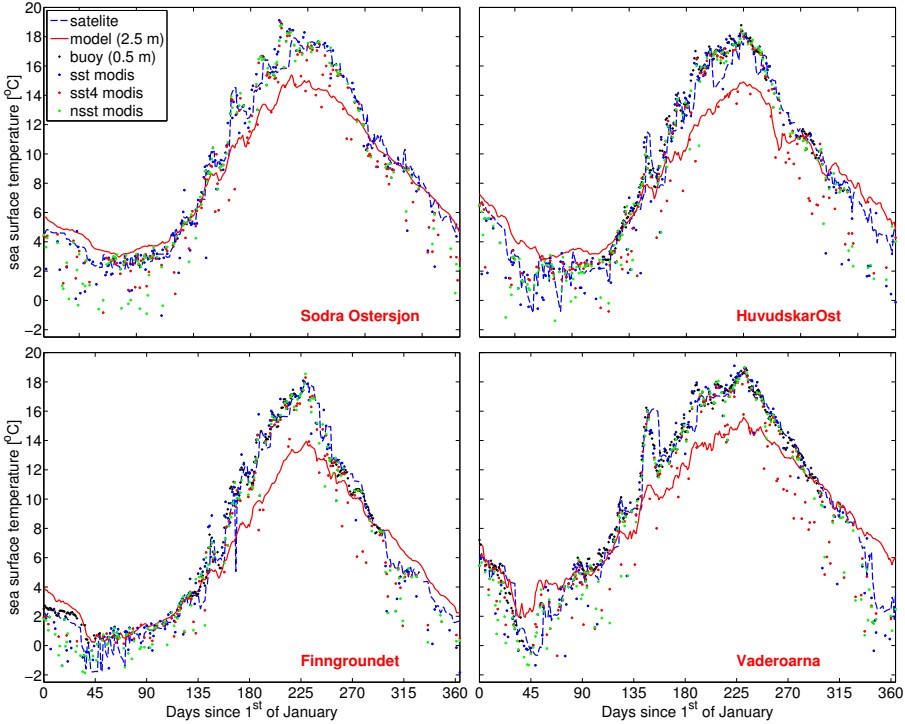

**Figure 8.** A comparison of the modeled and observed sea surface temperature at four different locations (based on daily data, results are for year 2012).

well reproduced by the model.



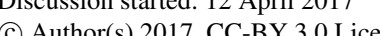



**Figure 9.** Temporal evolution of measured and modeled salinity (two upper panels) and temperature (two lower panels) profiles.



## 3   Ice model description and implementation for the Baltic Sea

The second active component in the presented Baltic Sea model is the Los Alamos sea ice model - Community Ice CodE
(CICE). CICE is a descendant of the basin-scale dynamic-thermodynamic and thickness distribution ice model (Hibler, 1979,
1980). The physics in the uncoupled ice model is the same as for the ice model used in the fully coupled system. The CICE

model is based on the energy conserving thermodynamics of (Bitz & Lipscomb , 1999). It contains multiple layers in each
thickness category and takes into account the effects of brine pockets within the sea ice. The ice dynamics utilizes the elastic-
viscous-plastic (EVP) rheology of (Hunke & Dukowicz, 1997, 2002). Sea ice ridging follows Rothrock (1975) and Thorndike
et al. (1975). CICE has several interacting components: a thermodynamic model that computes local growth rates of snow and
ice due to vertical conductive, radiative and turbulent fluxes, along with snowfall; a model of ice dynamics, which predicts the

velocity field of the ice pack based on a model of the material strength of the ice; a transport model that describes advection
of the a real concentration, ice volumes and other state variables; and a ridging parameterization that transfers ice among
thickness categories based on energetic balances and rates of strain. The CICE also has multiple thickness categories and ice
thickness distribution evolves in time (CICE Scientific Reference). CICE has been successfully used in many regional and
global adaptations (for example Rinke et al. (2003); McLaren et al. (2006); Dzierzbicka & Jakacki (2013); Herman et al.

15   (2011)).

The model does not require a spin up because sea ice disappears in the Baltic Sea every summer. Horizontal resolution of 1/48
degree (about 2.3 km) is the same as in the ocean model. It has been configured for 5 vertical categories. The time step in the
ice model of 600 seconds is the same as the coupler time step.

### 3.1   Fluxes at the ice-ocean and atmosphere-ice interfaces

Generally, heat and momentum fluxes, which influence the sea ice are similar to the ocean-atmosphere fluxes at the open ocean
surface. The main difference is due to the ice albedo and ice area. The net energy flux from atmosphere to ice is calculated
based on the heat budget:

$$Q_{net} = i_o \left[ (1 - \alpha) Q_{sw} + Q_{lw} + Q_{lat} + Q_{sen} \right] \tag{11}$$

where $i_o$ is ice fraction and $\alpha$ is ice albedo.

Momentum flux required by ice model is based on the nonlinear integral boundary-layer theories (Hibler & Bryan, 1987).
Ocean and air momentum fluxes are calculated with the following equations:

$$\tau_a = \rho_a C_a |\mathbf{U}_g| (\mathbf{U}_g \cdot \cos \phi + \mathbf{k} \times \mathbf{U}_g \cdot \sin \phi) \tag{12}$$

$$\tau_w = \rho_w C_w |\mathbf{U}_w - \mathbf{u}| ((\mathbf{U}_w - \mathbf{u}) \cdot \cos \theta + \mathbf{k} \times (\mathbf{U}_w - \mathbf{u}) \cdot \sin \theta) \tag{13}$$





where **u** represents the sea ice velocity, $\mathbf{U}_g$ is the geostrophic wind velocity (assumed to be higher than the ice motion velocity), $\mathbf{U}_w$ is the ocean surface velocity, $c_a$ and $c_w$ are air and water drag coefficients, $\rho_a$ and $\rho_w$ are air and ocean density and $phi$ and $\theta$ are air and water turning angles.

## 3.2 Ice model validation and results

Sea ice is the key element of the whole Baltic Sea system. Numerous satellite products exist that can be used to validate the Baltic Sea model but only few are suitable for sea ice validation. Under the MyOcean framework (currently known as the Copernicus Marine Environment Monitoring Service) two sea ice datasets are available – the first from the Danish Meteorological Institute (DM) and the second provided by Finnish Meteorological Institute (FMI). The FMI data include sea ice concentration and thickness over the entire Baltic Sea and are available as daily data with the horizontal resolution of 1 km.

DMI provides only sea ice concentration with 2 km resolution. Both observational data sets are based on SAR images. The operational sea ice service at FMI provides ice parameters, produced on a daily basis during the Baltic Sea ice season. The ice thickness chart (ITC) is a product based on the most recent available ice chart (IC) and a SAR image. The DMI data are also based on the operational product.

Figure 10 shows a comparison between the modeled and the measured ice thickness and concentration for two different points,

mostly covered by ice. The first one (Fig. 10, left panel) is located in the Bothnian Bay ( 24.2944$^o$E, 65.2023$^o$N), while the second one lays in the Gulf of Finland (28.6988$^o$E, 60.1510$^o$N). It is clearly visible that variability of ice concentration and thickness provided as operational products is much lower than modeled values. The cause of it is unknown, yet it is suspected that SAR fails to sense minor changes in ice thickness and concentration. Even if the SAR data do not fully resolve the short-term variability, the ranges of measured and modeled sea ice concentration and thickness are similar as well as their changes

on longer times scales.



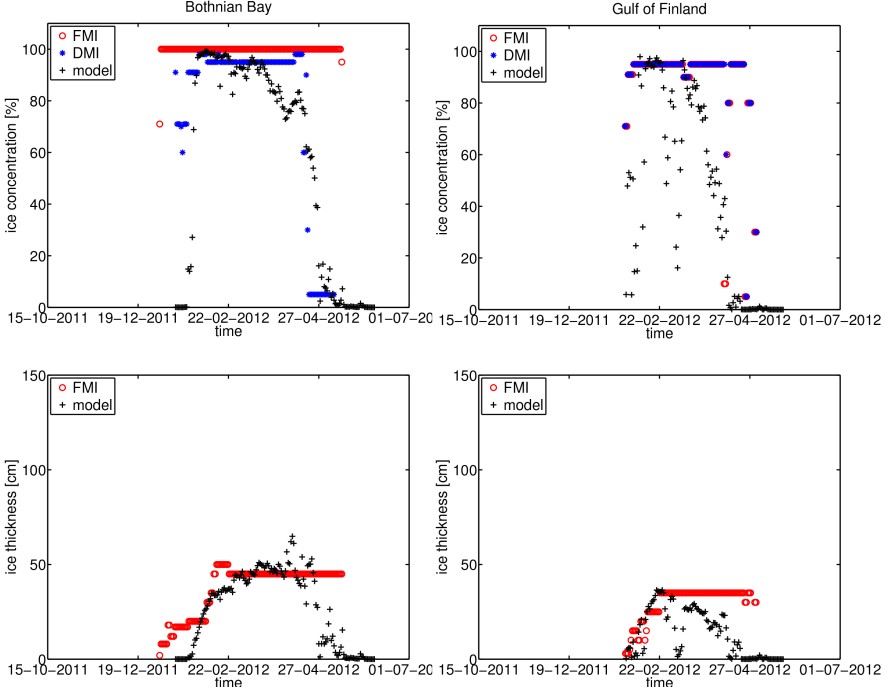

**Figure 10.** Comparison of the measured (FMI, DMI) and modeled ice concentration and thickness for the Bothnia Bay (left) and the Gulf of Finland (right).

Since daily ice charts (IC), produced by the Finnish Ice Service during each ice season, are available only as images, it is difficult to compare them with model results. To obtain more reliable validation, the sea ice thickness reproduced by the B-CESM model was compared to a combined product of SAR data and thermodynamic ice model HIGHTSI (HIGH-resolution Thermodynamic Snow/Ice model). HIGHTSI is a coupled snow and sea ice one-dimensional physical model targeted
5 to determine the snow/ice surface temperature, in-snow/ice temperature and snow/ice thickness at a selected site (Karvonen et al., 2003, 2007, 2008). Figures 11 and 12 present a comparison of sea ice thickness simulated with the presented B-CESM model with available ice charts based on SAR data combined with the thermodynamical ice model. The modeled and derived from SAR/HIGHTSI mean level ice thickness do not match ideally but spatial patterns are similar in both cases. In particular a distribution and shapes of the open water areas are well reproduced by the Baltic Sea model. Ice thickness in the Gulf of
10 Finland is in the range between 10 and 30 cm in both cases. In the Gulf of Bothnia sea ice is approximately 15 cm thicker than the value obtained from the SAR/HIGHTSI data. Sea ice thickness is similar in both cases also in the Gulf of Riga.



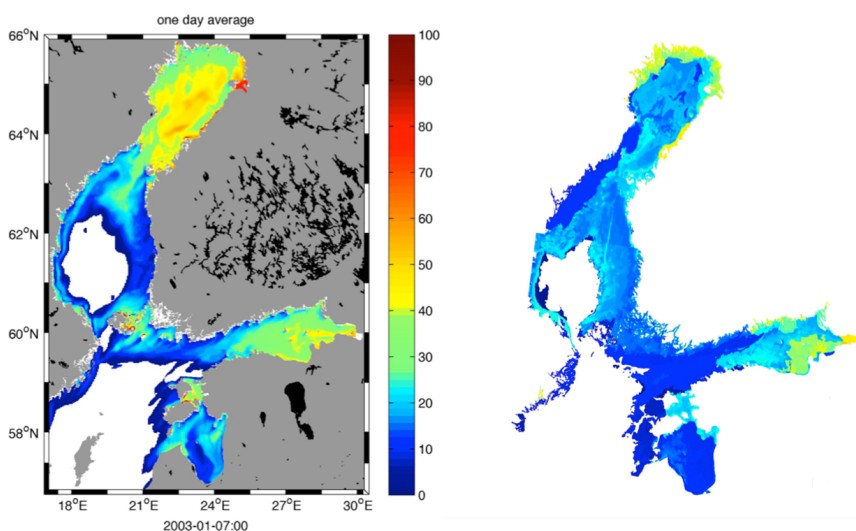

**Figure 11.** . Modeled (left) and measured (SAR) mean level ice thickness for the most of the Baltic Sea area (06-07.01.2003, color scales represent ice thickness in cm).

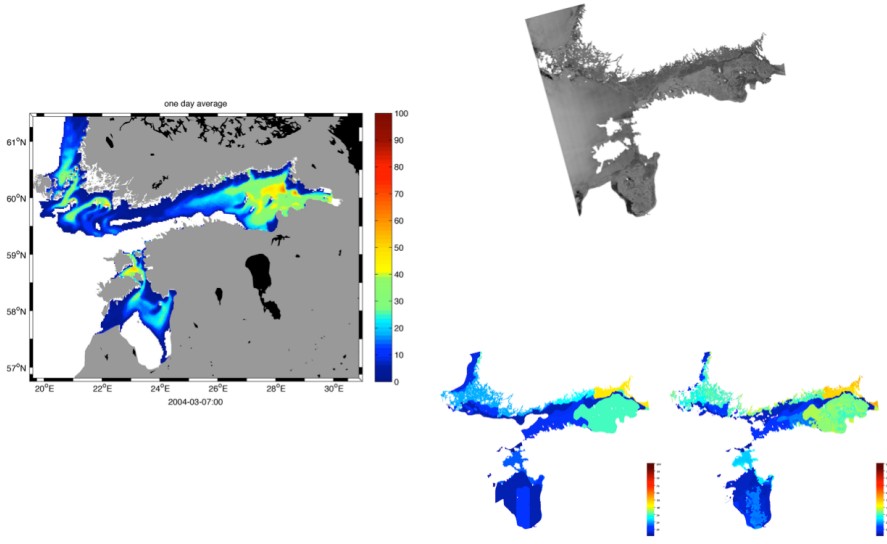

**Figure 12.** Mean level ice thickness generation process based on the data from SAR (three right images) and modeled ice thickness for the same day (07.03.2004, color scale represents ice thickness in cm).

To obtain an optimum balance between validation of sea ice parameters in the single location (precise but not very reliable) and validation for the sub-basins of the Baltic Sea (reliable but not sufficiently precise), in the next step we compare sea ice





parameters averaged in the selected boxes, representative for different parts of the Baltic Sea. Locations of the rectangles are shown on Fig. 13.

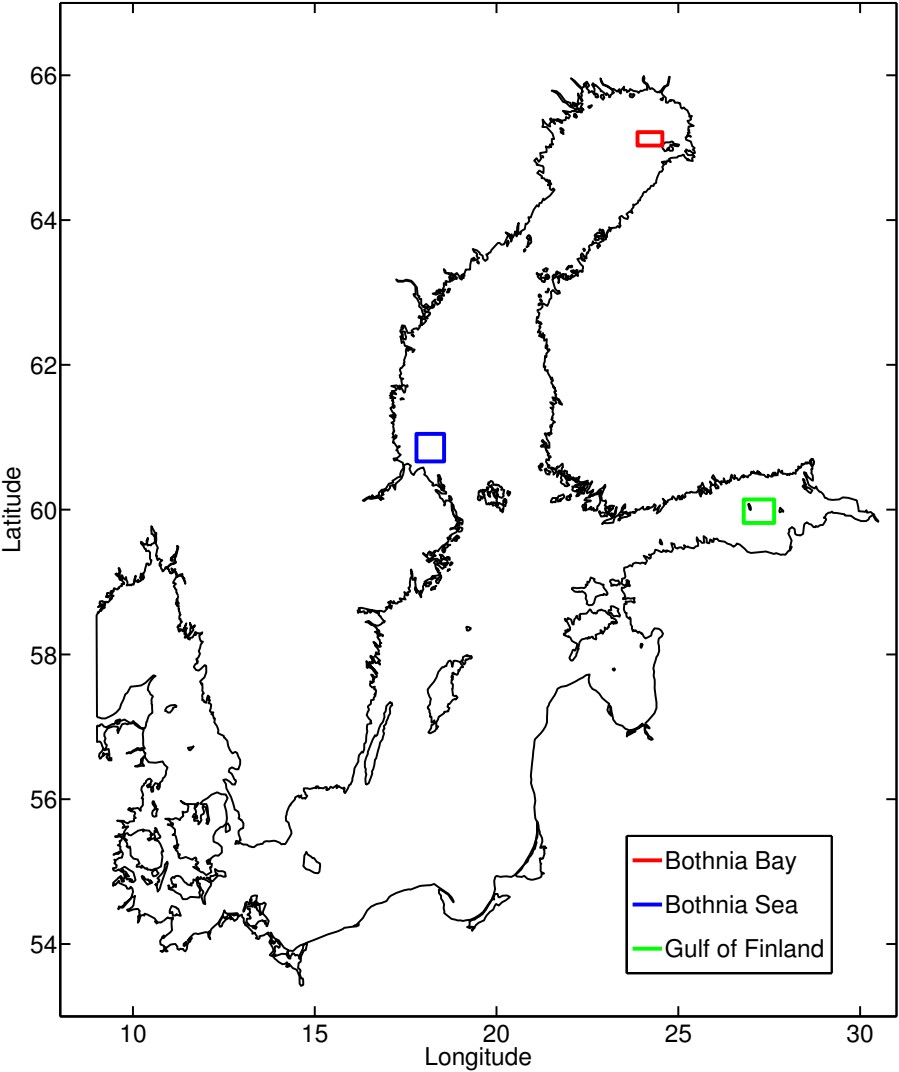

**Figure 13.** Locations of rectangles taken into account for validation of the ice cover.

The selection of boxes is based on the Climatological Ice Atlas for the Baltic Sea (CIABS , 1982). Ice frequency, average ice thickness and probability of ice occurrence within three ranges of thickness (0-20 cm, 20-50 cm and thicker than 50 cm) are shown on Fig. 14 for three selected boxes in the Bothnian Bay, the Bothnian Sea and the Gulf of Finland. Each panel of Fig. 14 shows a comparison of three model simulations (A, B and C) with climatological data obtained for each box from CIABS and data from Swedish Ice Service (SIS) provided by SMHI. It is important to add, that simulations and data cover different time periods. Data from CIABS are provided for years 1963-1979, SIS are for time period 2007-2013. The model simulation A was



for 9 km horizontal resolution and was forced by ERA40 reanalysis. Simulations B and C are identical except external forcing. B was forced by ERA40 and C was forced by ERA Interim (both are reanalysis provided by ECMWF).

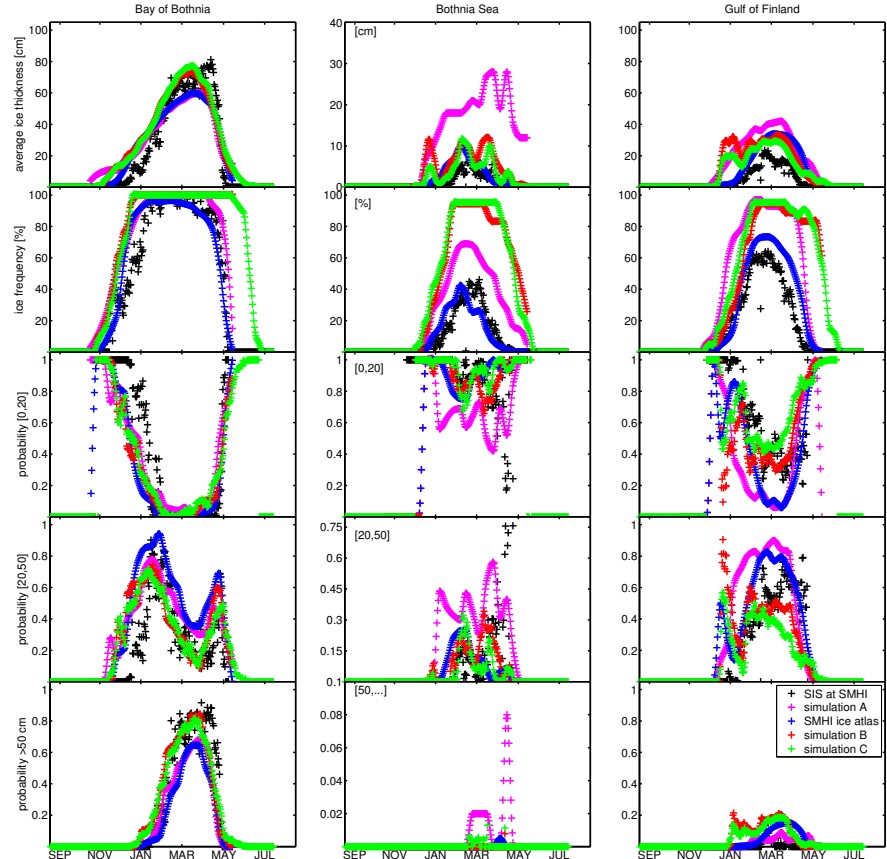

**Figure 14.** Ice frequency, average ice thickness and probability that ice thickness is less then 20 cm, between 20 and 50 cm and above 50 cm for three boxes indicated on Fig. 13 (the Bothnia Bay, the Bothnia Sea and the Gulf of Finland shown column-wise from left to right).

The best agreement between the modeled and measured sea ice variables was obtained in the box located in the Bothnian Bay (Fig. 14, left panels). Ice frequency and averaged ice thickness from the simulation A almost ideally agrees with climatological

5 data, while in the second and third simulations (B and C) reproduced ice thickness slightly higher than observed. Furthermore, there is a very good agreement between the measured and the modeled probabilities of sea ice occurrence in different thickness classes. The shapes of probabilities reflect the natural variability of sea ice thickness. In the beginning of winter thin ice appears. Then, ice thickness increases, which is visible in higher probabilities of ice thickness in the range of 20-50 centimeters. After some time lag ice thickness reaches 50 centimeters and more. These time shifts are visible in the probability distributions.

10 Towards the end of winter melting processes result in decreasing thickness of sea ice cover. It is reflected in a visible symmetry (quasi-symmetry) of probability that ice thickness is in the range of 0-20 cm.

In the beginning of November sea ice appears in the northern part of the Bothnian Bay. Then, the freezing process spreads



southwards. The range of values and temporal evolution of sea ice frequency and average ice thickness are similar in all boxes. The average ice thickness in the Bothnian Sea simulated by the model is lower than obtained from climatology. We think, the differences in the ice probability shown for Bothnia Sea and Gulf of Finland suggest there could be problem with ice dynamics. Ice cover in Bothnia Bay is much stable and influence of sea currents is smallest comparing to Bothnia Sea and Gulf of Finland.

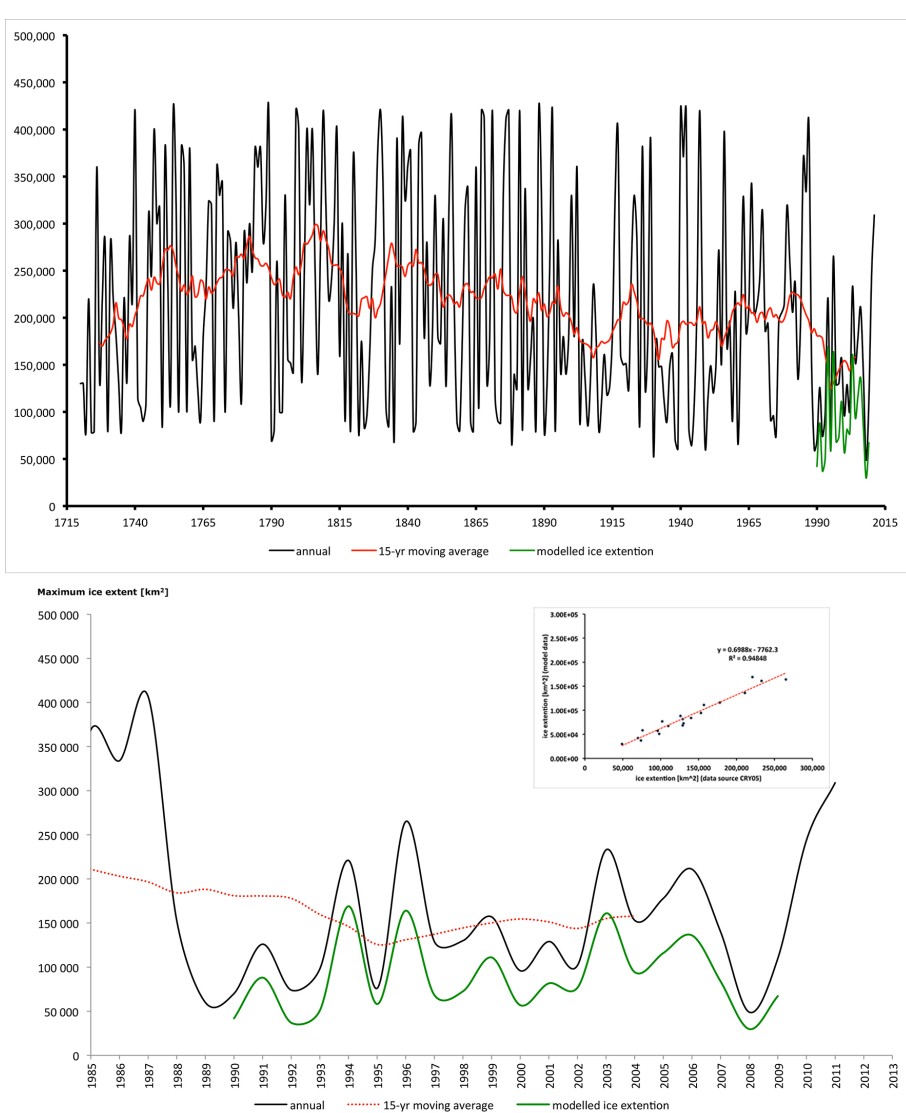

**Figure 15.** A) Maximum annual extent of ice cover in the Baltic Sea since 1720 Seinä & Palosuo (1996); Seinä et al. (2001, 2006). B) Maximum annual sea ice extent in the Baltic Sea in the last three decades from observations and B-CESM (and correlation between modeled and observational data as an insert).



Long term variability of ice cover in the Baltic Sea is shown on Fig. 15A as maximum annual sea ice extent since the 18th century (Seinä & Palosuo (1996); Seinä et al. (2001, 2006)). The black line depicts yearly values, the red line shows 15-year moving average and the modeled maximum sea ice extent has been overlaid for the last two decades (green line).

The maximum Baltic Sea annual ice extent has been slowly decreasing for the last two centuries. After a strong drop observed
in the beginning of 1990s due to unknown reasons, the sea ice maximum extent has been increasing again. Figure 15 B shows the detailed comparison between modeled and climatological sea ice extent for this period. The linear regression between modeled and measured data provides the correlation coefficient higher than 0.9, as shown by the insert on Fig. 15B.
Furthermore, an increase in the ice cover is also visible in the last three decades. The main factor that could cause a visible positive trend of ice extent is temperature. However, global warming has been observed in recent decades and it could suggest
that the Baltic Sea ice extent should have a negative tendency. Cohen et. al. (2014) shown that for the years 1990-2013 DJF surface temperature trends (per decade) are negative or neutral over the Baltic Sea area (Fig. 16) which could explain the trend.

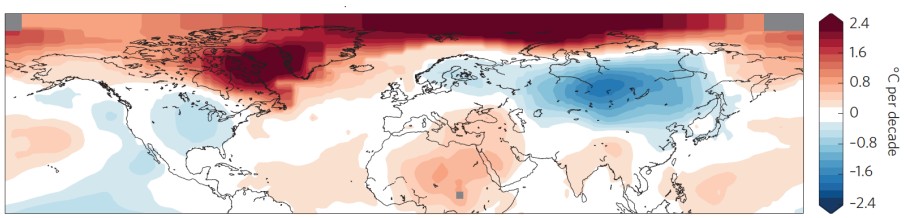

**Figure 16.** Winter temperature trends for the most recent period from 1990 (Cohen et. al., 2014).

Another, much less important factor that could have influence on the ice extent is fresh water from the catchment area. The Baltic Sea runoff is an important part of the fresh water budget. In general, freshwater flux minus evaporation is positive and has
a strong influence on the Baltic Sea circulation. Meier (2003) showed that the mean winter runoff to the Baltic Sea (January-February-March (JFM)) has been increasing since around 1970. The influence of runoff on the ice extent is much lower then heat budget, but excessive runoff decreases sea surface salinity. It is well known that freezing temperature depends on salinity and is going down when salinity increases (the lowest freezing temperature of the oceans is about $-1.8^oC$. Additional fresh water amount can decrease salinity which is equivalent to increasing freezing temperature and as a consequence it could
accelerate growth of the ice extent

## 4 Summary

High resolution coupled ice-ocean model has been successfully developed and implemented for the small regional domain of the Baltic Sea area. The main modification has been done in the ocean component which represents the lower boundary layer of the ice component. The model is based on a new open-source product - CESM. Both components (ice and ocean) were
validated and the validation confirms quite good agreement with the in-situ measurements. Furthermore, evidence confirming



increased ice extent was shown for the last two decades. It is an impact of the negative trend of winter air temperature since 1990 over the Baltic Sea are and also a consequence of increasing winter fresh water discharge in this period which results in the rise of freezing temperature, which in turn increases the ice extent. The main problem that appeared in the model is lower oscillations of the sea level, which is the result of linear approximation of free surface parameterization. In the future it

5   is planned to implement another free surface scheme (probably nonlinear explicit), which will fix the problem.



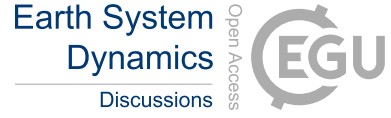

**Table A1.** Vertical resolution

| Model level | Tickness [cm] | middepth[cm] | Lower depth [cm] |
|---|---|---|---|
| 1 | 500 | 250 | 500 |
| 2 | 500 | 750 | 1000 |
| . . . | 500 | 1250 | 1500 |
| 50 | 500 | 24750 | 25000 |
| 51 | 500 | 25250 | 25500 |
| 52 | 603 | 25802 | 26103 |
| 53 | 728 | 26467 | 26103 |
| 54 | 878 | 27270 | 27709 |
| 55 | 1059 | 28239 | 28768 |
| 56 | 1277 | 29407 | 30045 |
| 57 | 1540 | 30815 | 31585 |
| 58 | 1858 | 32514 | 33443 |
| 59 | 2241 | 34564 | 35684 |
| 60 | 2704 | 37036 | 38388 |
| 61 | 3262 | 40019 | 41650 |
| 62 | 3934 | 43617 | 45584 |
| 63 | 4746 | 47957 | 50330 |
| 64 | 5725 | 53193 | 56055 |
| 65 | 6906 | 59508 | 62961 |
| 66 | 8330 | 67126 | 71291 |

**Appendix A**



## Appendix B: Figures

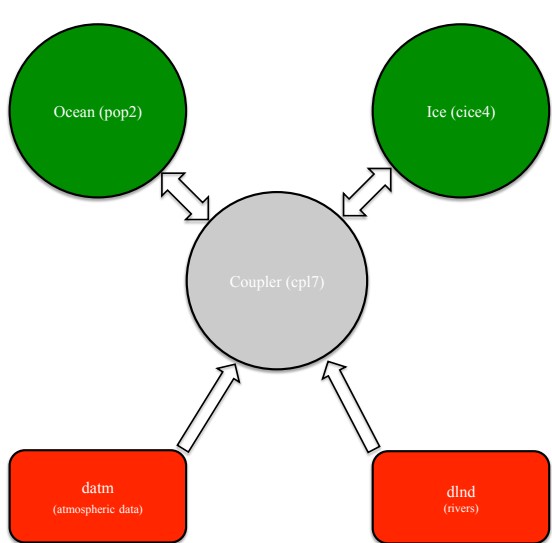

**Figure B1.** The Baltic Sea coupled model diagram



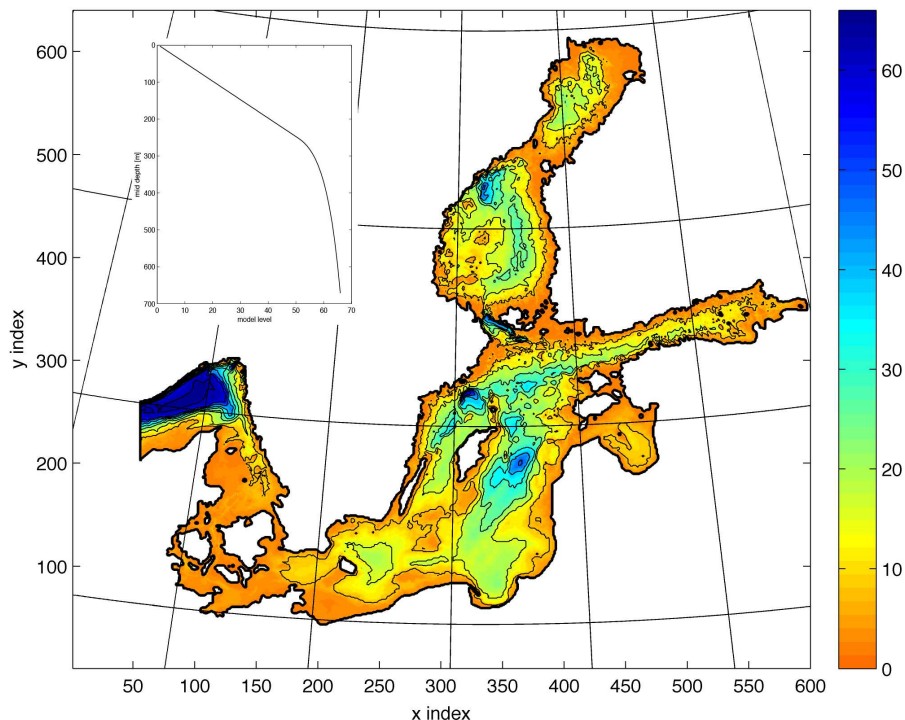

**Figure B2.** Model domain, bathymetry (color scale in model levels) and vertical resolution (insert).



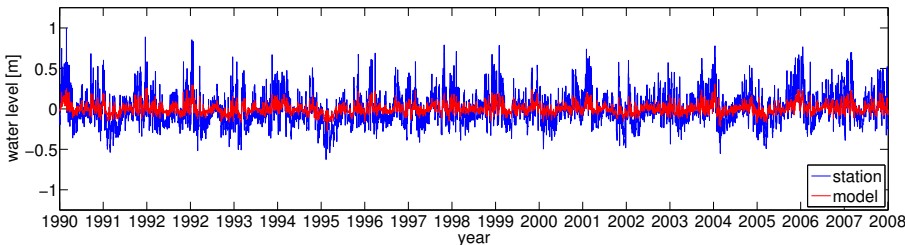

**Figure B3.** Modeled (red) and measured (blue) sea level in Göteborg.



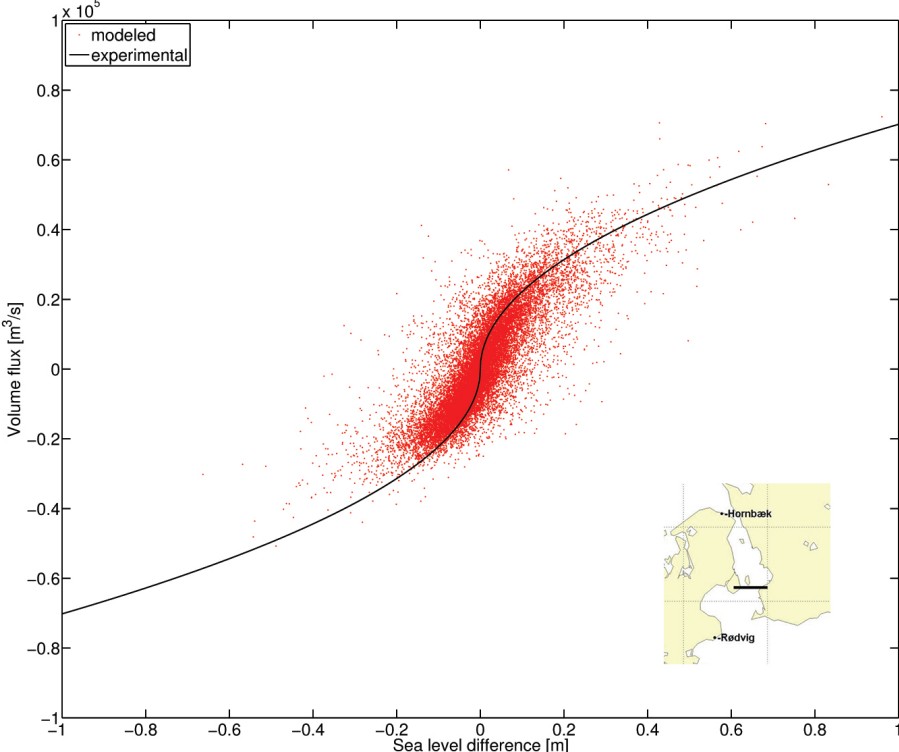

**Figure B4.** Flow rate vs. sea level difference in the Sund (red points – modeled, black line - calculated from the measured sea level difference).

The insert shows position of two stations at the northern and southern end of Sund and the section.





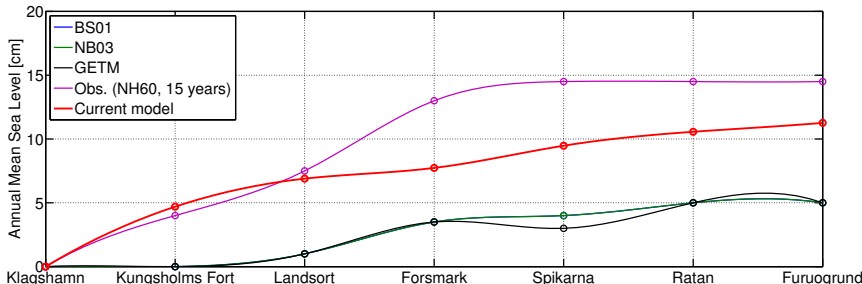

**Figure B5.** Comparison of the adjusted annual mean sea level from three different models, observations and current Baltic Sea model.




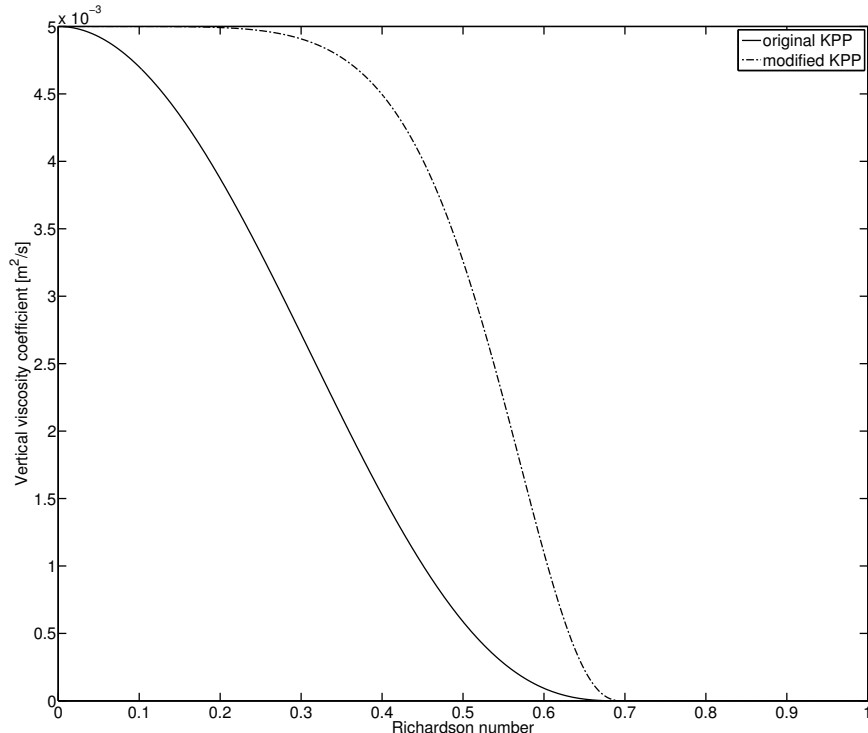

**Figure B6.** Original and modified (used in this simulations) turbulent viscosity (after (Durski et al. , 2004)).





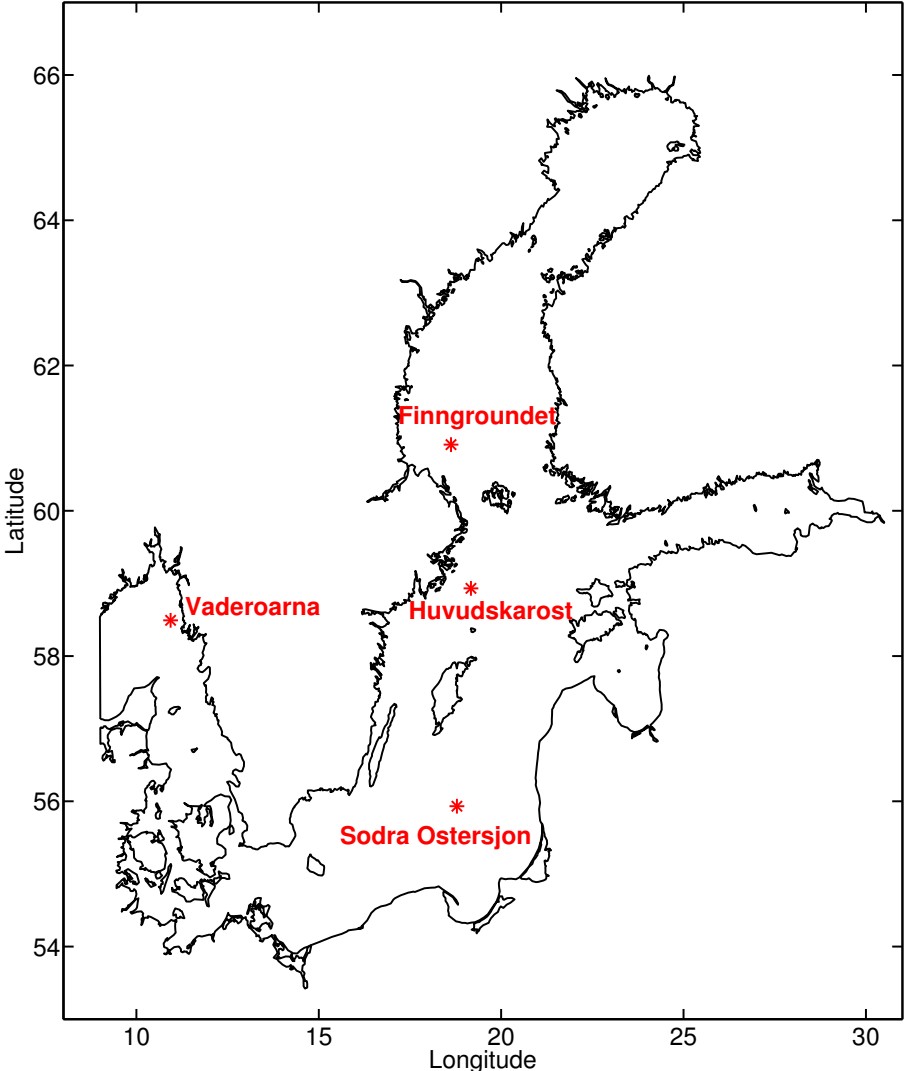

**Figure B7.** Locations of the buoys used for the comparison of the sea surface temperature.



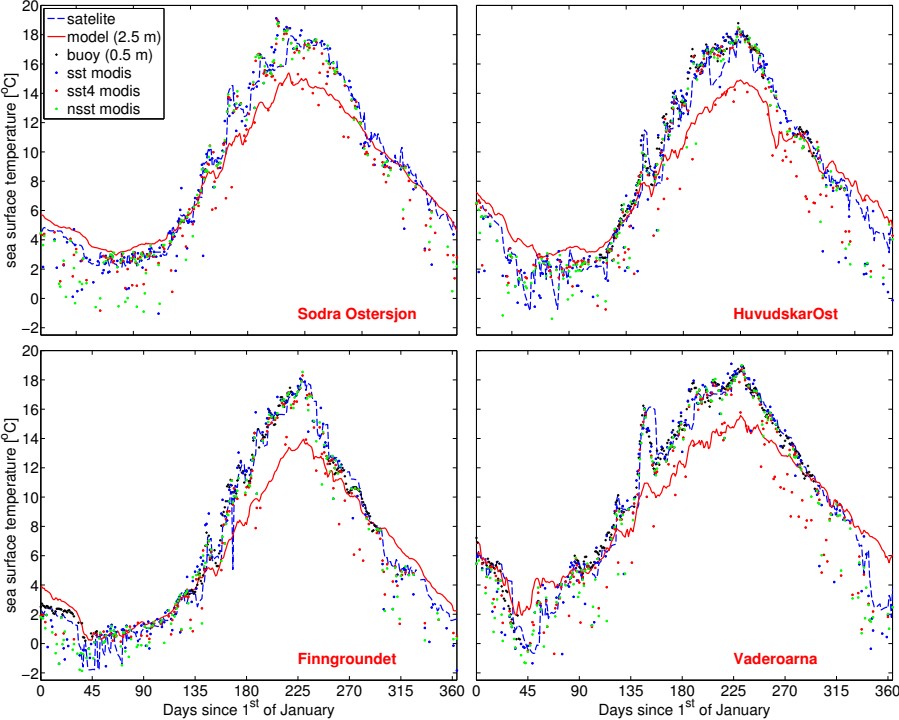

**Figure B8.** A comparison of the modeled and observed sea surface temperature at four different locations (based on daily data, results are for year 2012).



**Figure B9.** Temporal evolution of measured and modeled salinity (two upper panels) and temperature (two lower panels) profiles.





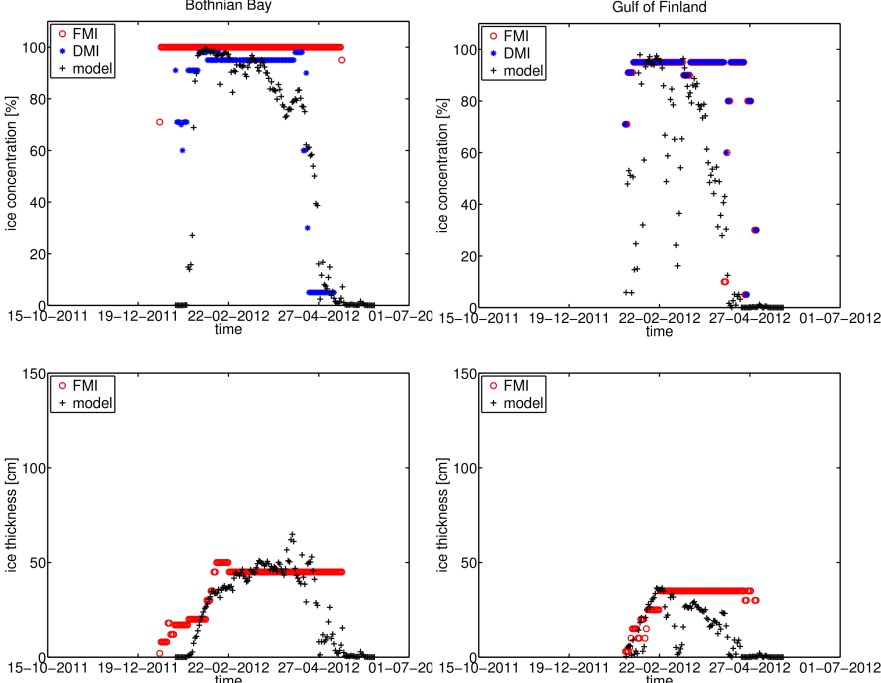

**Figure B10.** Comparison of the measured (FMI, DMI) and modeled ice concentration and thickness for the Bothnia Bay (left) and the Gulf of Finland (right).


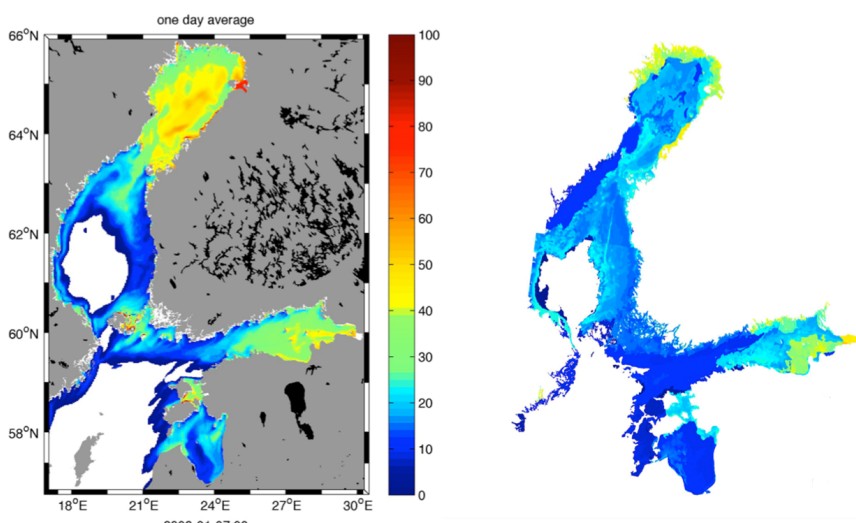

**Figure B11.** . Modeled (left) and measured (SAR) mean level ice thickness for the most of the Baltic Sea area (06-07.01.2003, color scales represent ice thickness in cm).



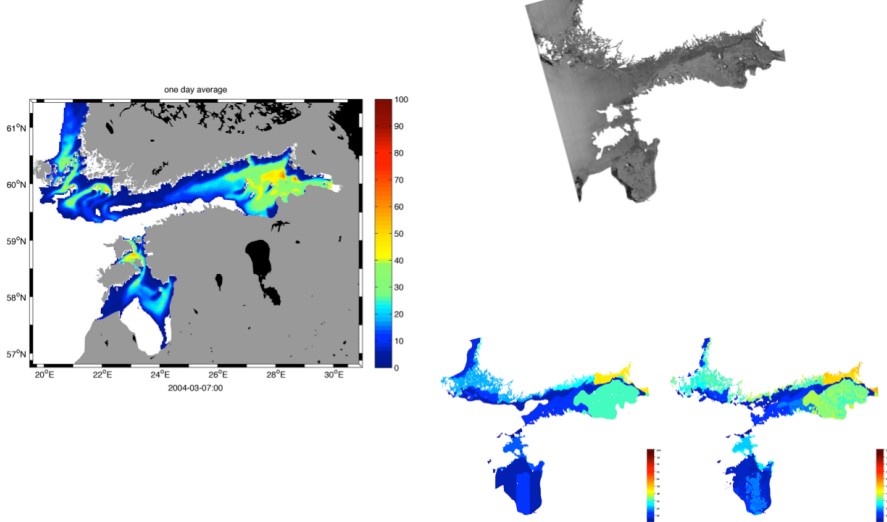

**Figure B12.** Mean level ice thickness generation process based on the data from SAR (three right images) and modeled ice thickness for the same day (07.03.2004, color scale represents ice thickness in cm).





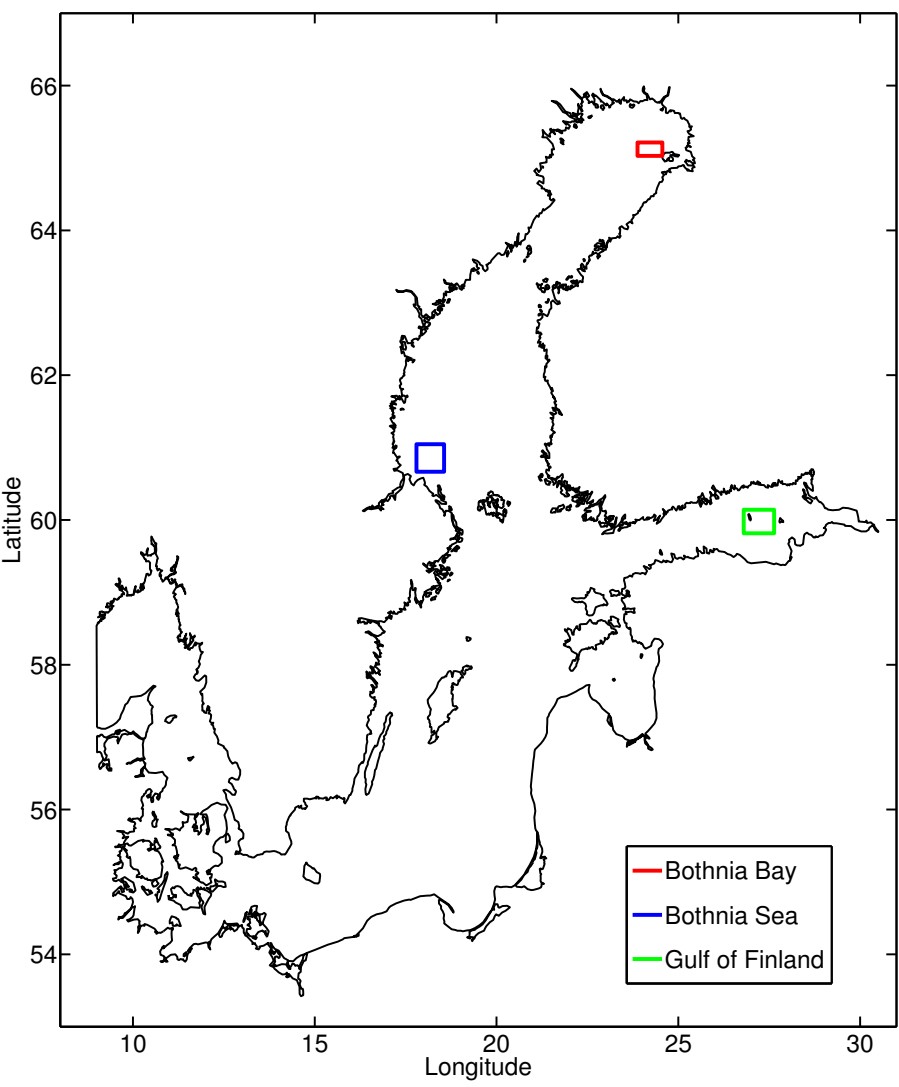

**Figure B13.** Locations of rectangles taken into account for validation of the ice cover.





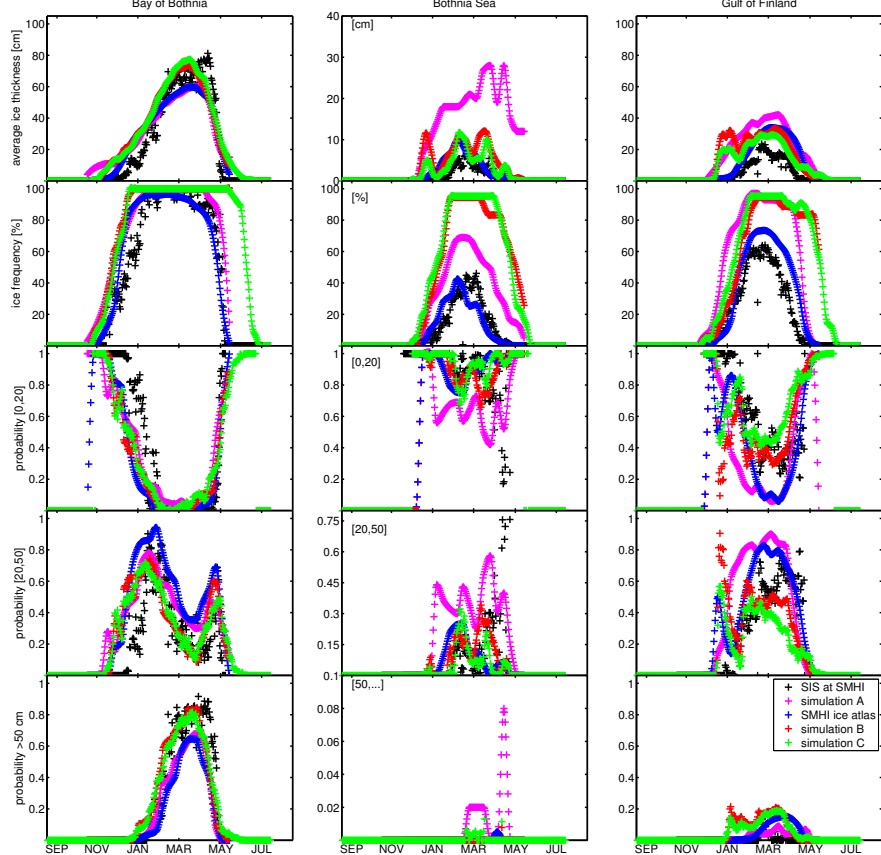

**Figure B14.** Ice frequency, average ice thickness and probability that ice thickness is less then 20 cm, between 20 and 50 cm and above 50 cm for three boxes indicated on Fig. 13 (the Bothnia Bay, the Bothnia Sea and the Gulf of Finland shown column-wise from left to right).



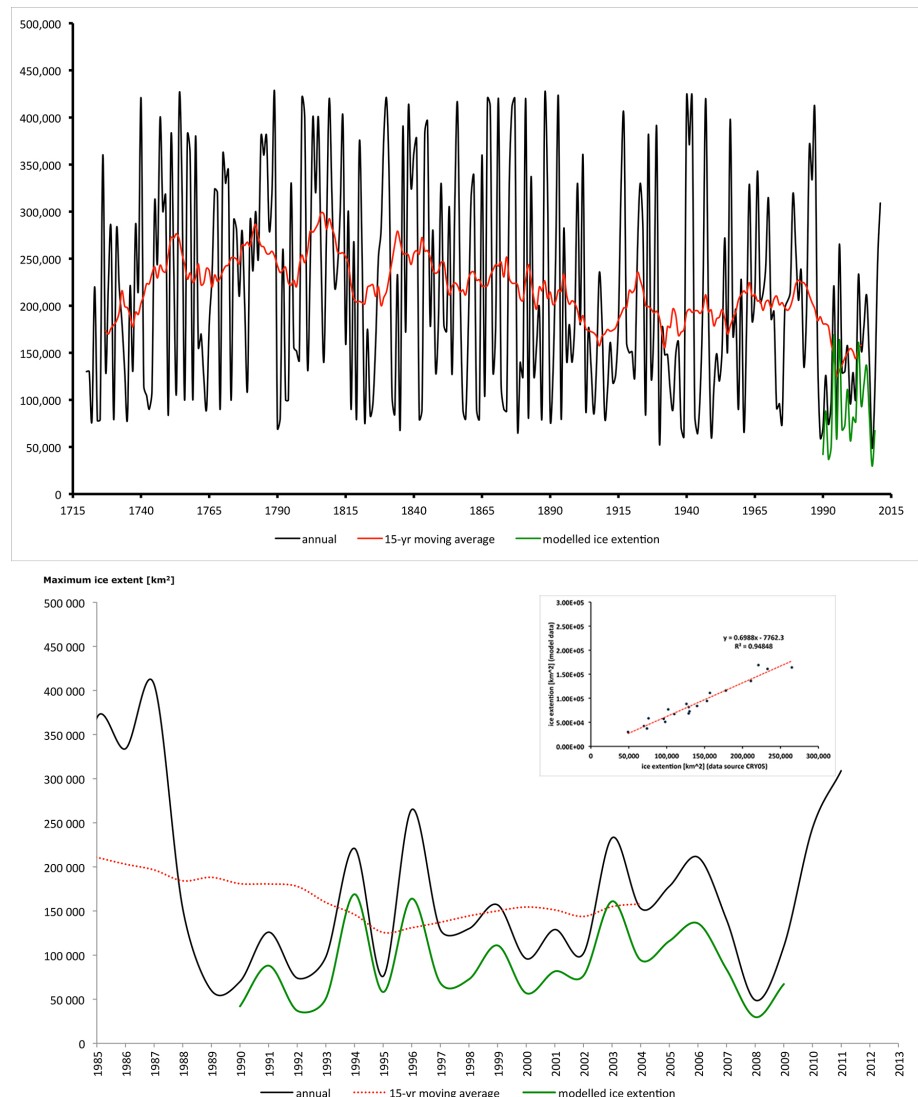

**Figure B15.** A) Maximum annual extent of ice cover in the Baltic Sea since 1720 (Seinä & Palosuo (1996); Seinä et al. (2001); Seinä et al. (2006)). B) Maximum annual sea ice extent in the Baltic Sea in the last three decades from observations and B-CESM (and correlation between modeled and observational data as an insert).





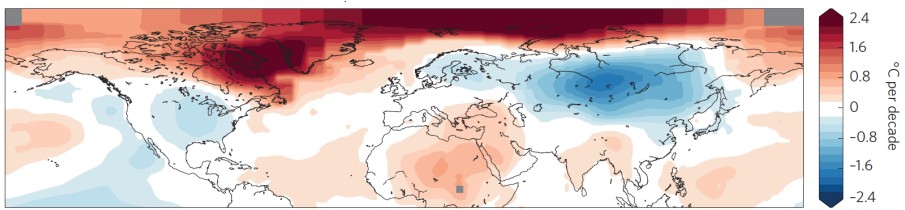

**Figure B16.** Winter temperature trends for the most recent period from 1990 (Cohen et. al., 2014).



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
