# Peer review of "An Evaluation and Implementation of the Regional Coupled Ice-Ocean Model of the Baltic Sea."

_Earth System Dynamics, 2017_

## Referee Comment (RC1) · Anonymous Referee #1 · 12 May 2017

The paper is about a coupled ice-ocean model the authors call B-CESM. It is based on the model CESM and the authors have made some modifications of the model and have set it up for the Baltic Sea. The paper includes model description and validation of the setup.

This paper does not investigate any scientific question.

I believe the paper would fit better in another EGU journal, namely Geoscientific Model Development (GMD).

The paper also needs more work both on the content and the presentation. The language needs to be polished too, once the content is OK.

[Figure]

My recommendation is reject, partly because of the misfit of the journal subject but also because of the main points one and two below. I must say that the shortcomings in the presentation also plays a role. They give the impression that the authors do not really care about their own manuscript. It could of course be that they have been under hard time pressure to submit due to project deadlines (if so they have my sympathy, haven't we all been there?).

I do recommend the authors to polish up the paper and resubmit to GMD instead. I hope my comments below will be of help in that process.

Main points:

1. The validations are shown in figures and the conclusion is always that the model is good (or occasionally that the observations are bad). It would be better to calculate some measures to get numbers on how good it is. It could be anything from RMS Error and correlation to different averages.

2. Many of the validations seem to use only one year of data even though the model has been run for much longer period. The outcome of the validation might depend strongly on which year was selected. Consider how to include a longer period in the validation. For illustration purposes, individual years can still be shown in figures.

3. Figure 9. If observations are this sparse, then find other observations. One option is to download data from www.ices.dk, which contain monthly data from many locations. I would recommend to use some of the more frequently visited locations such as BY15. This collection of data is lacking in quality control of the data so some care is advised.

4. Figure 10. If the observations are of too poor quality to be useful for the validation, then why include them at all in the paper?

5. Page 22 line 7. A correlation is good to include but it says nothing about the clearly visible bias. The bias needs to be quantified and commented upon.

6. Figure 15 A) does not add anything to the validation. Remove.

7. Page 22 line 8-20. This does not add anything to the validation. It is a discussion about a feature seen both in the model result and the observations but does not use the model to explain the phenomenon but is rather speculative to its nature. Remove.

Below are minor points or points about presentation.

The figure captions are not very descriptive and important information is lacking (even though present in the main text) to interpret the figure. One example is Figure 9 where the location is not stated in the caption.

Many figures have text that is too small, especially if the size will be reduced for the final publication.

There is a mismatch between headings and the text following the heading in the early part of the paper. Model description starts already in the introduction. Some very brief overview of the model could fit in the introduction but here are implementation details discussed, e.g., page 3, line 1-9 where river runoff implementation particular for this setup is discussed but lacking in the model description section. Further on, the text following the heading "Initial state of the model" starts with grid description and bathymetry.

Page 2, line 3, The name in the reference should be Arheimer (final r is missing).

On page 2, line 26 the model name and its abbreviation POP is introduced and used thereafter. On page 4 line 19, the full name is used as if it was the first time mentioned.

Page 3, line 2-4. Why would this method require removal of water at the boundary? Normally boundary conditions would take care of this, or in this case, the assimilation of sea level close to the boundary and no explicit removal would be necessary.

On page 3, line 10-12 it is stated that forcing was taken from two models. Please clarify if this was used for different runs or if they were combined in some way.

Page 4, line 28-31. I guess there should be a reference to Table A1 somewhere here.

Page 5, line 1-4: Are there no reference for the reanalysis?

As I understand the model domain, the boundary is a western boundary in Skagerrak as shown in Figure 2. On page 6 line 22-23 as well as on page 7 line 5-6, there are claims that there is a northern boundary. Please clarify where the boundary is and make sure the description is consistent with this.

The name Göteborg is sometimes spelled Goteborg, e.g., page 6 line7.

The location of Göteborg should be shown on a map, preferably Figure 2. The location of Göteborg relative to the boundary is of interest.

Page 7, line13. The name Sund is the Polish name for Øresund or the Sound. Please use established English names. Also Langeland Belt is the narrow part of the Greater Belt. I believe it is more common to talk about the latter in this context.

Page 7 line 15. The term sea level pressure is normally used for the atmospheric pressure while it here should be interpreted as the extra pressure from the water above the models rigid lid (possibly combined with the atmospheric pressure). I find this terminology confusing and too close to mean sea level pressure used to describe the forcing on page 4. Earlier, page 6, line 11, it was referred to as the surface pressure. I am not familiar with what the correct terminology would be, but please introduce it and be consistent with it.

Page 7, line 22, it is said that the black line in Figure 4 "represents measurements". What does this mean? Is it a theoretical model fitted to some measurements? Please clarify.

Page 9, line 2. I guess ERA40 interim should be ERA interim.

Page 9, line 1-3, I do not understand what this data have been used for. Is it used as forcing for different runs with B-CESM? If so, why was new runs needed? (Why) was it run with different resolution as implied in the text?

Page 9, line 3-4. What is meant by this sentence? Time averaging of course reduces the time variations but Figure 5 does not show time variations. Would it also reduce space variations? Why?

In section 2.4, the name of the buoy stations should be Väderöarna, Finngrundet, Huvudskär ost and Södra Östersjön.

Figure 7. Good with a map to show the location of the buoys but this might have been included in one map together with other locations and geographical names used in the paper.

Page 11, line 7. Observations are rarely complete. No need to state the obvious. What is more important is to state how the missing data is handled. For the figures shown it might be apparent, but if other kinds of analysis are done, as I suggest elsewhere, then this might become an issue.

page 12, line 4. The figure reference should be Fig 7.

Figure 8: It is hard to see the difference in colour on the small dots, especially the dark blue and black ones. Consider using symbols instead if bigger colour differences are not possible. It might be that there are too many dots for symbols to make sense and it just becomes impossible to see anything. Well, I point out the problem, the authors have to find the solution!

Page 15, line 17. What is vertical categories? Is it ice thickness categories as mentioned earlier in the text? Please clarify.

Formula (12) and (13). Right hand side are vector valued, so should the left hand side be and should therefore be boldface.

Page 16, line 2, $c_a$ and $c_w$ should be $C_a$ and $C_w$, respectively. Also, "phi" should probably be the Greek letter phi.

Page 16, line 8. The Danish Meteorological Institute is abbreviated DMI.

[Figure]

Page 16. I don't get a clear picture of the observations used here. Are there no references where I can read more about these? If not, better explanation is needed.

Page 16, line 15-16. Include these positions in some map together with other locations used in the paper.

Figure 10. Using dashes in the dates suggests ISO 8601 date format while the order of the numbers clearly indicates something else.

Figure 11 and 12. Why are the model and the observations shown so differently (land details included in one but not the other, frame and coordinates on one but not the other)?

Figure 12. What are the three different subfigures of observations? The numbers on the scale are too small for me to read. Is it the same scale for all subfigures including the model? I looks like there is a scale tick in the middle of the yellow on the small ones but not on the model scale.

Figure 13 can be included in a general figure with geographical locations.

Page 19, line 1. What is ice frequency? Please explain.

Page 19-20. The word probability belongs in probability theory. Here statistics are presented. It should probably(!) be called something with frequency or distribution. An observed frequency can be used to estimate a probability but it is not a probability by itself.

Page 19 line 8 - page 20 line 2. Why are several model runs used here? For which period is the model data that is used here? Does the 9 km resolution only apply to the ice model or also the ocean component of the model?

Page 19-20 The different time periods of the observations and the model is a problem due to the climate trend as well as big variation between ice seasons. I suggest to only use model data and observations for the same period of time.

Figure 14. The symbols are so thick and big, and at the same time so dense that it is hard to see the graphs of the different models and observations.

Figure 15 B). The moving average adds nothing here. Remove.

---

## Referee Comment (RC2) · Anonymous Referee #2 · 14 May 2017

The setup of a three dimensional coupled ice-ocean model for the Baltic Sea is described and some evaluation results for sea surface temperature, sea ice concentration and sea ice thickness for the period 1990-2009 are presented. The focus of the study is on the sea ice component. However, I miss a specific scientific question in the manuscript that could be addressed with the help of the introduced model. The manuscript resembles more a hasty written technical report than a peer-reviewed scientific article. Hence, I suggest to add a scientific analysis of the model results and to submit the manuscript later again, perhaps to ESD or any other journal that fits the considered, scientific question.

My comments in detail:

1) If the authors prefer to focus on sea ice changes in time, a trend analysis together with sensitivity experiments would be interesting to disentangle the drivers. The results of the sea ice model seem to be reasonably good for such an analysis (Fig. 14) although the sea ice extent is systematically underestimated (Fig. 15). However, for such a purpose the simulated interannual variations of sea ice extent, sea ice concentration and sea ice thickness need to be evaluated. Please compare with Löptien et al. (2013, Journal of Geophysical Research –Oceans).

2) The described ocean and sea ice models are based on a version of the well-known Bryan-Cox-Semtner ocean circulation model and the sea ice model CICE, respectively. These models have been applied to the Baltic Sea in many process and climate related studies since the 1990s (e.g. Lehmann 1995, Tellus; Meier et al. 2003; Journal of Geophysical Research –Oceans). Please introduce the relevant literature and compare your setup with the setups of previous studies. Is there anything novel in your model implementation and setup that may lead to improved results compared to the earlier studies?

3) Please evaluate both the seasonal cycle and the interannual variability of model variables in more statistical means (e.g. Eilola et al., 2011, Journal of Marine Systems). For instance, Figures 11 and 12 do not allow any conclusion on model quality. For the evaluation of SSTs satellite data are not well suited (except for spatial patterns) because model results (average of the upper 5 m) and skin temperatures are usually very different (Fig. 8). I suggest to use in addition to satellite data observations from the national monitoring programs of the various Baltic Sea countries collected in the Baltic Environmental Data base (BED) or elsewhere.

4) It is rather unlikely that the recent trend in ice extent is explained by increasing winter runoff because the salinity in the northern Baltic Sea is rather low and the impact of changes in sea surface salinity on the freezing point temperature is negligible in that salinity range.

---

## Author Comment (AC1) · 15 May 2017

Dear Reviewer,

As you are shown, the manuscript does not fill ESD review criteria. Also it contains a lot of shortcomings. I will withdraw the manuscript, figure out problems you mentioned and will submit it to Geoscientific Model Development journal. Thank you for spending time for making this review and it could be nice to have pre-reviewer like you (I know it is impossible).

Respectfully

Jaromir Jakacki